# FedPAT: Federated Test-Time Adaptation via Prototype Affinity Topology

**Shunxin Guo** [* 1 2]  **Jiaqi Lv** [* 1 2]  **Zhiqiang Kou** [1 2]  **Shuxia Lin** [1 2]  **Xin Geng** [1 2]

## Abstract

Federated Learning (FL) enables privacy-preserving collaboration among distributed clients in open-world environments, but its performance often degrades under data heterogeneity and unpredictable distribution shifts. Test-Time Adaptation (TTA) has recently been introduced into FL to leverage unlabeled data from unseen clients for online adaptation. However, most existing federated TTA methods employ local feature statistics, which can be brittle under diverse and severe distribution shifts. In this work, we observe that despite significant variations in feature distributions, the relational structure among class prototypes—termed *prototype affinity topology*—remains remarkably stable across heterogeneous clients. Building on this insight, we propose FedPAT, a **Fed**erated TTA framework that leverages **P**rototype **A**ffinity **T**opology as a cross-client structural prior. FedPAT learns a global PAT by aggregating class prototypes from source clients, capturing consensus inter-class relationships that are robust to local distribution variations. For unseen target clients, we design a topology-aware mechanism that enhances predictions via diffusion of the global PAT, fuses them with parametric outputs, and performs lightweight optimization for robust test-time adaptation. Extensive experiments demonstrate that FedPAT consistently outperforms advanced federated TTA and classical TTA methods across various distribution shifts.

## 1. Introduction

Federated learning (FL) (McMahan et al., 2017) has emerged as a key paradigm for privacy-preserving collaborative model training, enabling distributed clients to jointly learn a shared global model without exposing their raw data. While most existing FL methods are developed and evaluated under controlled and relatively stable settings (Guo et al., 2024a), practical deployments are often far more open and dynamic. In such real-world scenarios, distribution shifts can arise unexpectedly due to diverse factors such as hardware variations, environmental noise, or unknown data acquisition conditions, thereby violating the implicit assumption of distributional stationarity underlying global model training. Under these conditions, even the well-optimized global FL model may exhibit pronounced performance degradation, particularly when encountering out-of-distribution inputs from previously unseen clients. [1]

To address performance degradation caused by distribution shifts, Test-Time Adaptation (TTA) (Wang et al., 2022; Liang et al., 2020) has been extended to the FL setting, giving rise to a line of approaches commonly referred to as federated TTA. By leveraging unlabeled test data from unseen clients, federated TTA enables online model adaptation at deployment time, thereby enhancing robustness and generalization under distribution shifts. Recent federated TTA methods have explored diverse adaptation strategies, such as adaptive module updates (Bao et al., 2023), output distribution alignment (Rajib et al., 2025), and feature-level collaboration (Zhang et al., 2024), demonstrating encouraging improvements under moderate distribution shifts.

However, most existing federated TTA approaches implicitly assume that test-time adaptation can be reliably driven by local statistics estimated from unlabeled target data, such as normalization parameters (Wang et al., 2021), prediction entropy (Liang et al., 2020), or pseudo-label distributions (Wang et al., 2022; Rajib et al., 2025). In open federated environments, this assumption is particularly fragile, as unseen clients may experience severe and heterogeneous distribution shifts, which often result in unstable feature representations and biased predictions. This challenge is further amplified in federated settings, where unseen clients

---

[1] School of Computer Science and Engineering, Southeast University, Nanjing 210096, China [2] Key Laboratory of New Generation Artificial Intelligence Technology and Its Interdisciplinary Applications, Ministry of Education, Nanjing, China . Correspondence to: Xin Geng <xgeng@seu.edu.cn>.

*Proceedings of the 43$^{rd}$ International Conference on Machine Learning*, Seoul, South Korea. PMLR 306, 2026. Copyright 2026 by the author(s).

[1]Appendix B.2.5 provides empirical evidence illustrating this phenomenon.

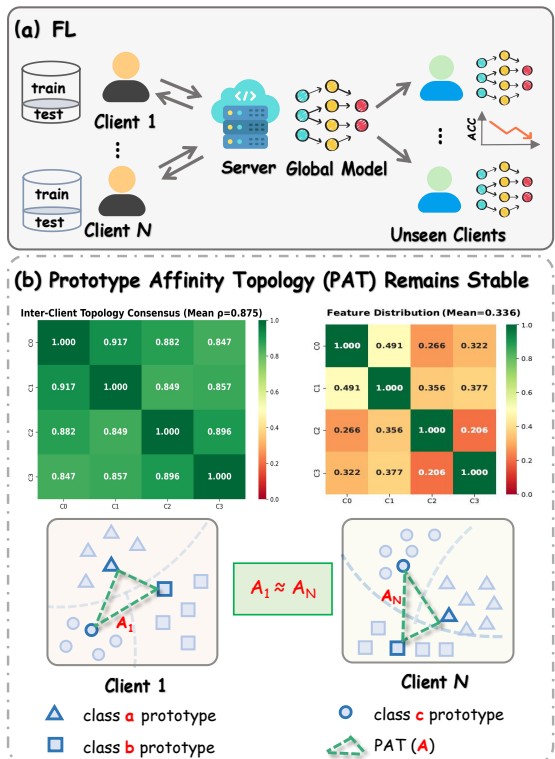

*Figure 1.* Motivation of FedPAT. **(a)** FL problem: the global model suffers performance degradation on unseen clients with distribution shifts. **(b)** Observation: Prototype Affinity Topology (PAT) remains stable across heterogeneous clients.

have no access to source data or cross-client feedback to validate or correct their local adaptation signals. These limitations motivate the search for a form of prior knowledge that remains consistent across heterogeneous clients and is robust to client-specific distribution shifts.

We observe that although distribution shifts significantly distort local feature representations across heterogeneous clients, the relational structure among class prototypes remains comparatively stable. This invariant relational structure captures pairwise class-level relationships in the learned feature space and is largely preserved across clients despite substantial variations in local data distributions. We refer to this structure as the *Prototype Affinity Topology* (PAT). As illustrated in Figure 1(b), the average cross-client similarity of PATs is measured as 0.875, substantially higher than that of the corresponding feature distributions (0.336) [2], indicating that class-level relational structures preserve shared semantic information under heterogeneous distribution shifts and can serve as a stable global prior for federated test-time adaptation.

Motivated by this insight, we propose **FedPAT**, a federated test-time adaptation framework that introduces a Prototype

---

[2]PAT similarity: Spearman correlation; feature distribution similarity: MMD. For details, please refer to Section 5.3.

Affinity Topology as a global structural prior. After federated pre-training, each source client computes class prototypes from its local training data. These prototypes are transmitted to the server and aggregated to construct a global PAT, which captures consensus inter-class relationships across heterogeneous clients. To further improve robustness while preserving communication efficiency, each source client performs lightweight validation-based refinement, where the global PAT is used to calibrate local prototypes and adapt batch normalization statistics. When deployed to unseen target clients, the global PAT serves as a stable inter-class semantic prior to construct a topology-aware non-parametric inference module. This module is fused with a confidence-guided lightweight classifier to form a unified posterior, enabling FedPAT to achieve effective adaptation under diverse distribution shifts while requiring only minimal parameter updates at test time. Extensive experiments on the CIFAR-10-C, CIFAR-100-C, and Tiny-ImageNet-C datasets demonstrate that FedPAT achieves significant and consistent improvements over state-of-the-art federated TTA baselines and classical traditional TTA methods. Our contributions are as follows:

- We systematically identify and formalize the cross-client stability of prototype affinity topology, revealing the invariant relational structure among classes under distribution shifts.
- We propose a federated test-time adaptation framework with a global structural prior, which complementarily integrates topology-aware nonparametric inference and parametric inference modules to achieve robust adaptation across heterogeneous clients.
- Beyond performance gains, we conduct extensive evaluations under diverse distribution shifts, providing empirical insights into the robustness and scalability of FedPAT.

## 2. Related Work

**Federated Learning.** Federated learning (FL) (Tamirisa et al., 2024; Kou et al., 2026; Guo et al., 2025) enables decentralized model training without sharing raw data, addressing privacy and security concerns in distributed systems. Foundational methods such as FedAvg and FedProx (Li et al., 2020) focus on robust aggregation under heterogeneous client data, while personalized FL approaches (Huang et al., 2021; Tan et al., 2022b; Liu et al., 2025), exploit inter-client similarity to improve client-specific performance. Despite their effectiveness, most FL methods (Qi et al., 2023; Huang et al., 2023) assume a stationary test distribution and primarily optimize training-time objectives, leaving distribution shifts at deployment time insufficiently addressed. This limitation becomes critical in open environments where client data distributions evolve after training.

**Test-Time Adaptation.** Test-time adaptation (TTA) (Chen et al., 2022; Liang et al., 2025b; Wang et al., 2025) improves

model robustness to distribution shifts by leveraging unlabeled test data during deployment. Entropy-based methods, such as Tent (Wang et al., 2021), perform online adaptation by minimizing prediction entropy through normalization layer updates, while CoTTA (Wang et al., 2022) improves stability via pseudo-labeling and selective parameter updates. Beyond parameter-level adaptation, recent work (You et al., 2025) explores correlation-based alignment that exploits feature correlation structures without extensive parameter updates. While effective in centralized settings, such approaches face challenges when applied to federated scenarios, including convergence instability under heterogeneous client distributions and high communication costs due to frequent cross-client synchronization.

**Federated Test-Time Adaptation.** Recent studies have explored the integration of FL and TTA to enable unsupervised adaptation in federated systems. Adaptive Test-Time Personalization (ATP) learns module-specific adaptation rates based on cross-client distribution shifts (Bao et al., 2023). FedTHE (Jiang & Lin, 2023) improves robustness through global and local classifier ensembling during test-time adaptation. FedTSA (Zhang et al., 2024) introduces a spatiotemporal correlation-based collaboration mechanism built on local feature representations, allowing clients with similar data distributions to enhance personalized model aggregation. FedSPL (Liang et al., 2025a) employs a teacher–student framework with contrastive learning and label selection at test time to improve adaptation to target distributions. FedCTTA (Rajib et al., 2025) addresses non-stationary distribution shifts by estimating collaboration relationships among clients using randomly generated noisy samples. Latte (Bao et al., 2025) further extends federated TTA to vision–language models through collaborative client memory.

Different from existing federated TTA methods that primarily rely on local client statistics adaptation, our approach explicitly models cross-client relationships through a global prototype affinity topology. This design exploits shared structural information across clients rather than local instance-level statistics, providing an alternative mechanism for federated test-time adaptation under heterogeneous distribution shifts.

## 3. Preliminaries

**Problem Setup.** Consider a federated learning system comprising $N$ source clients $\mathcal{S} = \{S_i\}_{i=1}^N$ and $M$ unseen target clients $\mathcal{U} = \{U_j\}_{j=1}^M$. Each source client $S_i$ possesses a labeled local dataset $\mathcal{D}_{S_i} = \{(\mathbf{x}_k^{S_i}, y_k^{S_i})\}_{k=1}^{n_i}$ with $n_i$ samples drawn from its local distribution $P_{S_i}(\mathbf{x}, y)$, where $\mathbf{x} \in \mathcal{X}$ denotes the input and $y \in \mathcal{Y} = \{1, \ldots, C\}$ its corresponding label from a shared label space of $C$ classes. In contrast, each unseen client $U_j$ has access only to an unlabeled test

dataset $\mathcal{X}_{U_j} = \{\mathbf{x}_k^{U_j}\}_{k=1}^{m_j}$ containing $m_j$ samples drawn from $P_{U_j}(\mathbf{x}, y)$, where the labels $\{y_k^{U_j}\}_{k=1}^{m_j}$ are unavailable during adaptation.

The local distributions $\{P_{S_i}\}_{i=1}^N$ and $\{P_{U_j}\}_{j=1}^M$ exhibit substantial heterogeneity due to variations in data collection procedures, device characteristics, and environmental conditions. We model this heterogeneity by assuming that all client distributions are sampled from a meta-distribution $\mathcal{Q}$ over probability distributions, i.e., $P_{S_i} \sim \mathcal{Q}, P_{U_j} \sim \mathcal{Q}$. Unseen clients $\{U_j\}_{j=1}^M$ do not participate in federated pre-training, and their distributions $\{P_{U_j}\}$ may differ substantially from source distributions due to data corruptions or novel deployment characteristics.

**Objective.** Our objective is to learn a model during federated pre-training that can be effectively adapted to unseen client distributions at test time, without access to labeled data or source client data. We formulate this as minimizing the expected adaptation risk over a distribution of clients:

$$\min_{\theta, \mathbf{A}_G} \mathbb{E}_{P \sim \mathcal{Q}} \left[ \mathcal{L}\big(f_{\mathcal{A}(\theta, \mathbf{A}_G, P)}; P\big) \right], \tag{1}$$

where $\theta$ denotes the model parameters and $\mathcal{A}(\theta, \mathbf{A}_G, P)$ is a test-time adaptation procedure that refines predictions using unlabeled samples from distribution $P$ guided by the $\mathbf{A}_G$.

## 4. Methodology

We introduce FedPAT, a federated test-time adaptation framework based on Prototype Affinity Topology (PAT). The overall pipeline is illustrated in Figure 2. We detail the construction of the global PAT, the lightweight validation refinement using a topology-guided objective, and the test-time adaptation mechanism for unseen target clients guided by the learned PAT.

### 4.1. Prototype Affinity Topology Construction

After multiple rounds of federated collaborative training (McMahan et al., 2017) to convergence, the pretrained global model $\mathbf{w}_G$ is distributed to each source client and used to initialize its local model $\mathbf{w}$.

**Local Prototype Computation.** For each source client $S_i$ with a labeled dataset $\mathcal{D}_{S_i} = \{(\mathbf{x}_k, y_k)\}_{k=1}^{n_i}$, we compute class prototypes by aggregating feature representations extracted by the pre-trained model's feature encoder $\phi(\cdot; \mathbf{w})$. The prototype for class $c \in \{1, \ldots, C\}$ is computed as:

$$\mathbf{p}_c^{S_i} = \frac{1}{|\mathcal{D}_{S_i}^c|} \sum_{(\mathbf{x}, y) \in \mathcal{D}_{S_i}^c} \phi(\mathbf{x}; \mathbf{w}), \tag{2}$$

where $\mathcal{D}_{S_i}^c = \{(\mathbf{x}, y) \in \mathcal{D}_{S_i} | y = c\}$ denotes the subset of samples belonging to class $c$. To facilitate cross-client comparison and ensure scale invariance, all prototypes are subsequently $\ell_2$-normalized.

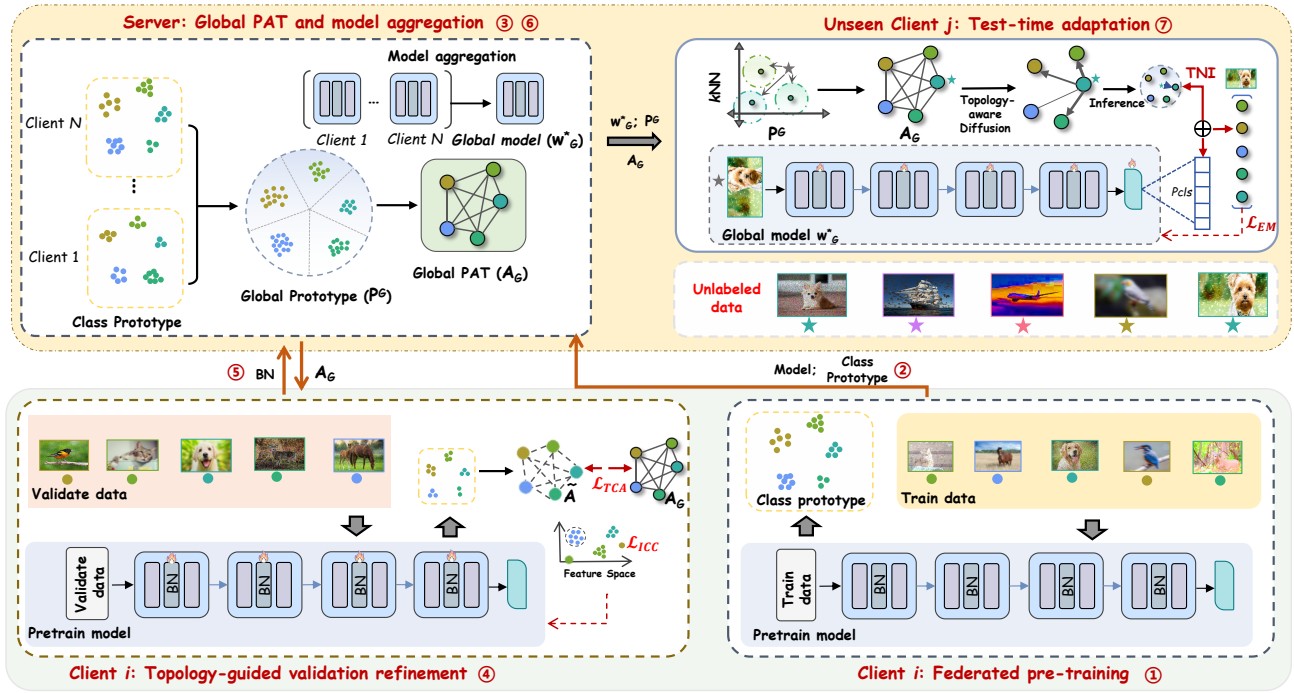

*Figure 2.* Overview of FedPAT. During federated pre-training, source clients collaboratively train a global model and extract class prototypes from their local training data. The server aggregates these prototypes to construct a global prototype affinity topology (PAT) $\mathbf{A}_G$ that encodes inter-class relationships. In the validation phase, source clients refine their models using topology-guided losses ($\mathcal{L}_{\text{TCA}}$ and $\mathcal{L}_{\text{ICC}}$) to align local prototype relationships with $\mathbf{A}_G$, and the server aggregates updated batch normalization statistics to produce $\mathbf{w}_G^*$. At test-time, unseen clients leverage $\mathbf{A}_G$ and global prototypes for topology-aware non-parametric inference, fuse these predictions with parametric outputs, and apply entropy minimization ($\mathcal{L}_{\text{EM}}$) to adapt to distribution shifts.

**Global Prototype Aggregation via Weighted Averaging.** The server constructs a global PAT $\mathbf{A}_G$ by aggregating class-wise prototypes from all participating clients. For each class $c$, local prototypes $\{\mathbf{p}_c^{S_i}\}$ are aggregated independently across clients to obtain a consensus global prototype $\mathbf{p}_c^G$. The global PAT is then constructed based on these aggregated class prototypes $\{\mathbf{p}_c^G\}_{c=1}^C$. This two-stage design explicitly mitigates feature space misalignment across heterogeneous clients, where local prototypes may reside in rotated or scaled feature spaces.

Each client $S_i$ computes and uploads its normalized class prototypes $\{\mathbf{p}_c^{S_i}\}_{c=1}^C$ to the server. For each class $c$, the server collects the set of local prototypes $\{\mathbf{p}_c^{S_i} | S_i \in \mathcal{S}\}$, and aggregates them via sample-weighted averaging to obtain the global prototype:

$$\mathbf{p}_c^G = \frac{\sum_{S_i \in \mathcal{S}} w_i^c \mathbf{p}_c^{S_i}}{\left\| \sum_{S_i \in \mathcal{S}} w_i^c \mathbf{p}_c^{S_i} \right\|_2}, \qquad (3)$$

where $w_i^c = n_i^c / \sum_{S_j \in \mathcal{S}} n_j^c$ weights clients by their number of $c$ class samples $n_i^c$. This weighted aggregation ensures that clients with more $c$ class samples contribute proportionally more to the global prototype, while $\ell_2$-normalization maintains unit sphere constraints.

**Global Affinity Topology Construction.** Given the global prototypes $\{\mathbf{p}_c^G\}_{c=1}^C$, the server constructs the global PAT $\mathbf{A}_G$ as a $C \times C$ affinity matrix encoding pairwise class relationships across the federation:

$$\mathbf{A}_G[c, c'] = \langle \mathbf{p}_c^G, \mathbf{p}_{c'}^G \rangle = \mathbf{p}_c^{G\top} \mathbf{p}_{c'}^G, \qquad (4)$$

where $\mathbf{A}_G[c, c'] \in [-1, 1]$ with values closer to 1 indicating higher affinity between classes. Our focus is on maintaining this global topology for consistent learning across clients, which can serve as a reference for model adaptation during validation and testing.

### 4.2. Topology-Guided Validation Refinement

For each source client $S_i$ with validation set $\mathcal{V}_{S_i}$, we refine the local model using a topology-guided validation objective composed of two complementary structural regularizers:

$$\mathcal{L}_{\text{VR}} = \lambda_{\text{TCA}} \mathcal{L}_{\text{TCA}} + \lambda_{\text{ICC}} \mathcal{L}_{\text{ICC}}, \qquad (5)$$

where $\mathcal{L}_{\text{TCA}}$ denotes the topology consistency alignment loss, $\mathcal{L}_{\text{ICC}}$ denotes the intra-class compactness loss, and $\lambda_{\text{TCA}}, \lambda_{\text{ICC}} > 0$ balance their contributions.

**Topology Consistency Alignment (TCA) Loss.** To align client-specific prototype relations with the global PAT $\mathbf{A}_G$, we minimize the structural discrepancy between batch-level

affinities and $\mathbf{A}_G$. For a mini-batch $\mathcal{B} \subseteq \mathcal{V}_{S_i}$, class-wise batch prototypes are computed as

$$\tilde{\mathbf{p}}_c = \frac{1}{|\mathcal{B}_c|} \sum_{(\mathbf{x},y) \in \mathcal{B}_c} \phi(\mathbf{x}; \mathbf{w}), \qquad (6)$$

where $\mathcal{B}_c = \{(\mathbf{x}, y) \in \mathcal{B} : y = c\}$. When a class is absent from the current batch, we substitute it with the corresponding local prototype. Each batch prototype $\tilde{\mathbf{p}}_c$ is $\ell_2$-normalized to unit norm before affinity computation. We then construct a batch affinity matrix, i.e., $\tilde{\mathbf{A}}[c, c'] = \langle \tilde{\mathbf{p}}_c, \tilde{\mathbf{p}}_{c'} \rangle$. The TCA loss measures the structural discrepancy using Smooth L1 loss on off-diagonal elements:

$$\mathcal{L}_{\text{TCA}} = \frac{1}{C(C-1)} \sum_{c \neq c'} \text{SmoothL1}\Big( \tilde{\mathbf{A}}[c, c'], \mathbf{A}_G[c, c'] \Big). \qquad (7)$$

This formulation ensures that both batch-level and global affinities are computed in the same normalized feature space, enabling meaningful structural alignment.

**Intra-Class Compactness (ICC) Loss.** To improve feature discriminability and reduce intra-class variance, we enforce compactness of feature representations within each class:

$$\mathcal{L}_{\text{ICC}} = \frac{1}{|\mathcal{C}_\mathcal{B}|} \sum_{c \in \mathcal{C}_\mathcal{B}} \frac{1}{|\mathcal{B}_c|} \sum_{(\mathbf{x},y) \in \mathcal{B}_c} \|\phi(\mathbf{x}; \mathbf{w}) - \tilde{\mathbf{p}}_c\|_2^2, \quad (8)$$

where $\mathcal{C}_\mathcal{B} = \{c \mid |\mathcal{B}_c| > 0\}$ denotes the set of class indices that appear in the current mini-batch $\mathcal{B}$. This loss regularizes the local feature space by pulling samples toward their corresponding class prototypes, thereby stabilizing prototype estimation under distribution shifts.

**Batch Normalization Adaptation and Aggregation.** Each client $S_i$ adapts its local Batch Normalization (BN) layers on the validation set $\mathcal{V}_{S_i}$ by optimizing the affine parameters $\boldsymbol{\gamma}_i = \{\gamma_i^{(l)}\}_{l=1}^L$ and $\boldsymbol{\beta}_i = \{\beta_i^{(l)}\}_{l=1}^L$ via gradient descent on the $\mathcal{L}_{\text{VR}}$ loss. During this process, BN statistics $\boldsymbol{\mu}_i = \{\mu_i^{(l)}\}_{l=1}^L$ and $\boldsymbol{\sigma}_i^2 = \{(\sigma_i^2)^{(l)}\}_{l=1}^L$ are also updated through forward propagation. The server then aggregates both the affine parameters and statistics:

$$\begin{aligned} \boldsymbol{\mu}_G &= \sum_{S_i \in \mathcal{S}} \omega_i \boldsymbol{\mu}_i, \quad \boldsymbol{\sigma}_G^2 = \sum_{S_i \in \mathcal{S}} \omega_i \boldsymbol{\sigma}_i^2, \\ \boldsymbol{\gamma}_G &= \sum_{S_i \in \mathcal{S}} \omega_i \boldsymbol{\gamma}_i, \quad \boldsymbol{\beta}_G = \sum_{S_i \in \mathcal{S}} \omega_i \boldsymbol{\beta}_i, \end{aligned} \qquad (9)$$

where $\omega_i = \frac{|\mathcal{V}_{S_i}|}{\sum_{j \in \mathcal{S}} |\mathcal{V}_{S_j}|}$ reflects the reliability of each client's validation set and can be used to aggregate the refined global model $\mathbf{w}_G^*$.

### 4.3. Topology-Consistent Test-Time Adaptation

At test-time deployment, an unseen client $U_j$ receives the trained global model $\mathbf{w}_G^*$ and global PAT $\mathbf{A}_G$, and global class prototypes $\{\mathbf{p}_c^G\}_{c=1}^C$.

**Topology-aware Non-parametric Inference (TNI).** TNI is a structure-aware inference module that refines non-parametric $k$NN predictions by propagating class-level posteriors over a learned global PAT, enabling robust test-time inference without gradient updates. Given a test feature representation $\mathbf{z} = \phi(\cdot; \mathbf{w}^*)$, we compute the cosine similarity $s_c = \langle \mathbf{z}, \mathbf{p}_c^G \rangle$ with each global prototype. The prototype-based posterior is obtained by selecting the top-$k$ classes with highest similarities and applying softmax:

$$p_{\text{proto}}(y = c | \mathbf{z}) = \begin{cases} \dfrac{\exp(s_c)}{\sum_{c' \in \mathcal{N}_k} \exp(s_{c'})}, & c \in \mathcal{N}_k(\mathbf{z}) \\ 0, & \text{otherwise} \end{cases} \qquad (10)$$

where $\mathcal{N}_k(\mathbf{z})$ denotes the top-$k$ prototype indices.

While $p_{\text{proto}}$ captures instance-level similarity, it ignores relationships among classes. To incorporate relationships among classes, we refine $p_{\text{proto}}$ by propagating it over the global $\mathbf{A}_G$. Before normalization, we transform the $\mathbf{A}_G$ into a non-negative graph and add self-loops to ensure well-defined random-walk dynamics and numerical stability. We then perform a topology-aware diffusion defined as:

$$p_{\text{TNI}} = (1 - \alpha) \, p_{\text{proto}} + \alpha \cdot \mathbf{D}^{-1} \mathbf{A}_G \, p_{\text{proto}}, \qquad (11)$$

where $\alpha$ controls the diffusion strength, $\mathbf{D} = \text{diag}(\mathbf{A}_G \mathbf{1})$ is the degree matrix, and $\mathbf{D}^{-1} \mathbf{A}_G$ represents the row-normalized transition matrix. This diffusion process propagates predictive mass to related classes, yielding a topology-consistent and robust non-parametric prediction.

---

**Theoretical Insight**

Topology-aware diffusion can be viewed as a single-step approximation to a graph-regularized optimization problem. The refined class posterior $\mathbf{p} \in \mathbb{R}^C$ approximately solves

$$\min_{\mathbf{p}} \ \|\mathbf{p} - p_{\text{proto}}\|_2^2 \ + \ \alpha \, \mathbf{p}^\top \mathbf{L}_{\text{rw}} \mathbf{p},$$

where $\mathbf{L}_{\text{rw}} = \mathbf{I} - \mathbf{D}^{-1} \mathbf{A}_G$ is the random-walk graph Laplacian induced by the global PAT (after adding self-loops and row normalization). The closed-form solution is $\mathbf{p}^* = (\mathbf{I} + \alpha \mathbf{L}_{\text{rw}})^{-1} p_{\text{proto}}$. Our diffusion step in Eq. 11 yields

$$p_{\text{TNI}} = (\mathbf{I} - \alpha \mathbf{L}_{\text{rw}}) \, p_{\text{proto}} \ \approx \ (\mathbf{I} + \alpha \mathbf{L}_{\text{rw}})^{-1} p_{\text{proto}},$$

which acts as a graph-smoothing regularizer and improves robustness under distribution shifts.

---

*Illustrative Example.* Consider a simple class graph where class $c_1$ exhibits higher affinity to $c_2$ than to $c_3$ in the global PAT. Given an initial prediction concentrated on $c_1$, topology-aware diffusion propagates probability mass toward $c_2$ through the graph structure, yielding a smoother

posterior aligned with inter-class relationships (Zhou et al., 2003; Zhu et al., 2003).

**Dual-Head Prediction with Confidence-Guided Fusion.** To balance robustness and discriminability, FedPAT employs a dual-head prediction strategy that combines the topology-aware non-parametric inference with a parametric classifier. The parametric prediction is obtained from the globally optimized classifier:

$$p_{\text{cls}} = \text{softmax}\left(\frac{h(\mathbf{z}; \mathbf{w}_G^*)}{\tau_{\text{cls}}}\right), \qquad (12)$$

where $h(\cdot; \mathbf{w}_G^*)$ denotes the linear classification head. While highly discriminative, this parametric predictor may be sensitive to test-time distribution shifts. Therefore, we design a consistency-aware adaptive fusion mechanism to combine both predictions. The confidence score is defined as follows:

$$c_{\text{cls}} = \max_c p_{\text{cls}}[c], \quad c_{\text{TNI}} = \max_c p_{\text{TNI}}[c] \qquad (13)$$

and derive a sample-wise adaptive gate $g = \sigma(c_{\text{cls}} - c_{\text{TNI}})$, where $\sigma(\cdot)$ is the sigmoid function. The fused class posterior is computed as:

$$p_{\text{final}} = g \cdot p_{\text{cls}} + (1 - g) \cdot p_{\text{TNI}}. \qquad (14)$$

This adaptive fusion can dynamically select a more reliable prediction probability vector during testing, improving robustness under different heterogeneous distribution shifts. To further reduce prediction uncertainty, we apply entropy minimization (Wang et al., 2021) on the fused posterior:

$$\mathcal{L}_{\text{EM}} = \frac{1}{|\mathcal{B}|} \sum_{\mathbf{x} \in \mathcal{B}} \mathcal{H}(p_{\text{final}}(\mathbf{x})), \qquad (15)$$

where $\mathcal{B} \subseteq \mathcal{X}_{U_j}$ denotes a test batch. This objective is applied only to lightweight components (the classifier head and normalization layers), and serves as a standard confidence refinement step.

# 5. Experiment

In this section, we design comprehensive experiments to answer the following three key research questions:

- **RQ1 (Effectiveness):** Can FedPAT outperform state-of-the-art federated TTA and classical TTA methods in diverse and severe distribution shift scenarios?
- **RQ2 (Interpretability):** Does the global Prototype Affinity Topology (PAT) capture meaningful structural knowledge across heterogeneous clients?
- **RQ3 (Component Analysis):** Which components of FedPAT are essential to its performance under extreme distribution shifts?

## 5.1. Experimental Setup

**Datasets and Federated Partitioning.** We evaluate FedPAT on three standard corruption benchmarks: CIFAR-10-C (Hendrycks & Dietterich, 2019), CIFAR-100-C (Hendrycks & Dietterich, 2019), and Tiny-ImageNet-C. Each dataset is constructed by adding 15 corruption types of varying severity to a base dataset. We employ a standard three-way split approach (Yuan et al., 2021), randomly splitting the dataset into 300 clients including 240 source clients and 60 unseen target clients.

**Distribution Shift Scenarios.** We evaluate model robustness under three distribution shift scenarios: *Feature Shift (FS):* Following (Bao et al., 2023), source clients are pre-trained with 15 corruption types, then tested on 4 unseen corruptions at severity level 5, evaluating generalization to novel feature perturbations. *Label Shift (LS):* Each client exhibits a long-tailed class distribution, creating heterogeneous label distributions. *Hybrid Shift (HS):* Combines both shifts in which source clients have imbalanced label distributions and feature corruptions, while target clients face entirely new label distributions and unseen corruption types. Detailed experimental settings is provided in Appendix B.1.

**Baseline Methods.** We compare with federated TTA methods including ATP (Bao et al., 2023), which learns adaptive update rates, FedCTTA (Rajib et al., 2025), which performs similarity-based noise aggregation, and FedSPL (Liang et al., 2025a), which employs a teacher-student framework with contrastive learning. BN-Adapt and Tent (Wang et al., 2021) for batch normalization adaptation, SHOT (Liang et al., 2020) for feature extractor adaptation, T3A (Iwasawa & Matsuo, 2021) for classifier adjustment, MEMO (Zhang et al., 2022) for augmentation-based adaptation, EM and BBSE (Lipton et al., 2018) for label shift correction, and Surgical (Lee et al., 2023) for selective block adaptation. Since retraining the model with client-specific label weights is impractical in federated learning, we leverage the estimated label distribution to calibrate the classifier outputs. All methods use the ResNet18 as the backbone network in the main experiment. Additional detailed experiments on the ResNet50 and ViT-B/16 networks are provided in Appendix B.2.3.

## 5.2. Comparison with TTA Methods (RQ1)

To evaluate the effectiveness of the proposed method, we conduct comprehensive comparisons between FedPAT and state-of-the-art federated TTA approaches, as well as classical TTA methods. Table 1 shows the test accuracy on unseen client data with the most severe level of corruption (severity = 5) under various distribution shift scenarios across three benchmark datasets. The "No Adapt" row indicates the performance of the global model after training when directly testing on unseen client data. Compared with the significant

*Table 1.* Test-time adaptation accuracy (%) on unseen clients with severity = 5 corruption across three distribution shift types. Best in **bold**, second best underlined.

| Method | CIFAR-10-C | | | | CIFAR-100-C | | | | Tiny-ImageNet-C | | | |
|---|---|---|---|---|---|---|---|---|---|---|---|---|
| | FS | LS | HS | Avg. | FS | LS | HS | Avg. | FS | LS | HS | Avg. |
| *General TTA Methods* | | | | | | | | | | | | |
| No Adapt | $59.80_{\pm 0.1}$ | $72.31_{\pm 0.3}$ | $55.69_{\pm 0.2}$ | $62.60_{\pm 0.2}$ | $33.22_{\pm 0.3}$ | $48.36_{\pm 0.1}$ | $30.54_{\pm 0.3}$ | $37.37_{\pm 0.1}$ | $33.40_{\pm 0.1}$ | $48.87_{\pm 0.1}$ | $30.79_{\pm 0.1}$ | $37.69_{\pm 0.1}$ |
| BN-Adapt | $63.48_{\pm 0.1}$ | $72.66_{\pm 0.2}$ | $56.53_{\pm 0.2}$ | $64.22_{\pm 0.2}$ | $33.40_{\pm 0.2}$ | $48.87_{\pm 0.2}$ | $30.79_{\pm 0.2}$ | $37.69_{\pm 0.2}$ | $36.16_{\pm 0.2}$ | $50.39_{\pm 0.2}$ | $35.00_{\pm 0.2}$ | $40.52_{\pm 0.2}$ |
| SHOT | $49.98_{\pm 0.3}$ | $31.34_{\pm 0.3}$ | $27.72_{\pm 0.3}$ | $36.35_{\pm 0.3}$ | $23.90_{\pm 0.3}$ | $28.88_{\pm 0.3}$ | $23.06_{\pm 0.3}$ | $25.28_{\pm 0.3}$ | $29.17_{\pm 0.3}$ | $34.15_{\pm 0.3}$ | $28.65_{\pm 0.3}$ | $30.66_{\pm 0.3}$ |
| Tent | $67.32_{\pm 0.2}$ | $49.05_{\pm 0.2}$ | $43.38_{\pm 0.2}$ | $53.25_{\pm 0.2}$ | $38.27_{\pm 0.2}$ | $41.50_{\pm 0.2}$ | $34.20_{\pm 0.2}$ | $37.99_{\pm 0.2}$ | $40.05_{\pm 0.2}$ | $42.85_{\pm 0.2}$ | $38.05_{\pm 0.2}$ | $40.32_{\pm 0.2}$ |
| T3A | $63.69_{\pm 0.2}$ | $71.35_{\pm 0.2}$ | $55.87_{\pm 0.2}$ | $63.64_{\pm 0.2}$ | $33.40_{\pm 0.2}$ | $48.88_{\pm 0.2}$ | $30.80_{\pm 0.2}$ | $37.69_{\pm 0.2}$ | $36.15_{\pm 0.2}$ | $50.37_{\pm 0.2}$ | $34.99_{\pm 0.2}$ | $40.50_{\pm 0.2}$ |
| MEMO | $66.29_{\pm 0.2}$ | $77.26_{\pm 0.2}$ | $60.52_{\pm 0.2}$ | $68.02_{\pm 0.2}$ | $35.14_{\pm 0.2}$ | $\underline{50.18}_{\pm 0.2}$ | $32.90_{\pm 0.2}$ | $\underline{40.07}_{\pm 0.2}$ | $40.43_{\pm 0.2}$ | $49.84_{\pm 0.2}$ | $38.12_{\pm 0.2}$ | $42.80_{\pm 0.2}$ |
| EM | $58.94_{\pm 0.2}$ | $\underline{80.61}_{\pm 0.1}$ | $60.28_{\pm 0.2}$ | $66.61_{\pm 0.2}$ | $30.57_{\pm 0.2}$ | $49.36_{\pm 0.2}$ | $29.29_{\pm 0.2}$ | $36.41_{\pm 0.2}$ | $34.50_{\pm 0.2}$ | $50.42_{\pm 0.2}$ | $33.82_{\pm 0.2}$ | $39.58_{\pm 0.2}$ |
| BBSE | $54.36_{\pm 0.2}$ | $79.08_{\pm 0.1}$ | $58.50_{\pm 0.2}$ | $63.98_{\pm 0.2}$ | $5.87_{\pm 0.3}$ | $32.84_{\pm 0.3}$ | $4.17_{\pm 0.3}$ | $14.29_{\pm 0.3}$ | $22.54_{\pm 0.3}$ | $45.72_{\pm 0.2}$ | $20.93_{\pm 0.3}$ | $29.73_{\pm 0.3}$ |
| Surgical | $61.07_{\pm 0.2}$ | $77.15_{\pm 0.1}$ | $58.48_{\pm 0.2}$ | $65.57_{\pm 0.2}$ | $31.71_{\pm 0.2}$ | $49.04_{\pm 0.3}$ | $29.81_{\pm 0.2}$ | $36.85_{\pm 0.2}$ | $35.51_{\pm 0.2}$ | $50.56_{\pm 0.2}$ | $34.27_{\pm 0.2}$ | $40.11_{\pm 0.2}$ |
| *Federated TTA Methods* | | | | | | | | | | | | |
| ATP | $\underline{67.40}_{\pm 0.2}$ | $79.36_{\pm 0.1}$ | $\underline{61.86}_{\pm 0.2}$ | $\underline{69.54}_{\pm 0.1}$ | $\mathbf{39.81}_{\pm 0.2}$ | $48.89_{\pm 0.2}$ | $31.44_{\pm 0.2}$ | $40.04_{\pm 0.2}$ | $\underline{41.04}_{\pm 0.2}$ | $\underline{50.57}_{\pm 0.2}$ | $40.28_{\pm 0.2}$ | $\underline{43.96}_{\pm 0.2}$ |
| FedSPL | $63.63_{\pm 0.3}$ | $74.06_{\pm 0.2}$ | $58.16_{\pm 0.2}$ | $65.28_{\pm 0.2}$ | $33.51_{\pm 0.2}$ | $49.86_{\pm 0.2}$ | $31.60_{\pm 0.2}$ | $38.32_{\pm 0.2}$ | $35.52_{\pm 0.3}$ | $48.93_{\pm 0.1}$ | $35.41_{\pm 0.2}$ | $39.95_{\pm 0.2}$ |
| FedCTTA | $67.24_{\pm 0.2}$ | $54.51_{\pm 0.2}$ | $47.14_{\pm 0.2}$ | $56.30_{\pm 0.2}$ | $35.62_{\pm 0.2}$ | $42.50_{\pm 0.2}$ | $\underline{33.48}_{\pm 0.2}$ | $37.20_{\pm 0.2}$ | $36.83_{\pm 0.2}$ | $42.07_{\pm 0.2}$ | $36.95_{\pm 0.2}$ | $38.62_{\pm 0.2}$ |
| **FedPAT** | $\mathbf{67.83}_{\pm 0.1}$ | $\mathbf{80.73}_{\pm 0.1}$ | $\mathbf{63.13}_{\pm 0.2}$ | $\mathbf{70.56}_{\pm 0.1}$ | $\underline{38.99}_{\pm 0.2}$ | $\mathbf{50.31}_{\pm 0.2}$ | $\mathbf{34.62}_{\pm 0.2}$ | $\mathbf{41.31}_{\pm 0.2}$ | $\mathbf{42.22}_{\pm 0.2}$ | $\mathbf{50.70}_{\pm 0.2}$ | $\mathbf{41.29}_{\pm 0.2}$ | $\mathbf{44.74}_{\pm 0.2}$ |

*Table 2.* Test-time adaptation accuracy (%) on Tiny-ImageNet-C with random corruption severity across various distribution shifts.

| Method | FS | LS | HS | Avg. |
|---|---|---|---|---|
| *General TTA Methods* | | | | |
| No Adapt | $36.57_{\pm 0.2}$ | $40.52_{\pm 0.2}$ | $35.09_{\pm 0.2}$ | $37.39_{\pm 0.2}$ |
| BN-Adapt | $36.58_{\pm 0.2}$ | $50.23_{\pm 0.2}$ | $35.11_{\pm 0.2}$ | $40.64_{\pm 0.2}$ |
| SHOT | $28.76_{\pm 0.3}$ | $33.66_{\pm 0.3}$ | $15.18_{\pm 0.3}$ | $25.87_{\pm 0.3}$ |
| Tent | $40.01_{\pm 0.2}$ | $42.89_{\pm 0.2}$ | $30.59_{\pm 0.2}$ | $37.83_{\pm 0.2}$ |
| T3A | $36.58_{\pm 0.2}$ | $50.31_{\pm 0.2}$ | $35.08_{\pm 0.2}$ | $40.66_{\pm 0.2}$ |
| MEMO | $39.11_{\pm 0.2}$ | $\underline{50.69}_{\pm 0.2}$ | $\underline{37.88}_{\pm 0.2}$ | $42.56_{\pm 0.2}$ |
| EM | $\underline{42.39}_{\pm 0.2}$ | $50.32_{\pm 0.2}$ | $34.27_{\pm 0.2}$ | $42.33_{\pm 0.2}$ |
| BBSE | $36.23_{\pm 0.3}$ | $46.07_{\pm 0.2}$ | $30.77_{\pm 0.3}$ | $37.69_{\pm 0.3}$ |
| Surgical | $35.88_{\pm 0.2}$ | $50.43_{\pm 0.2}$ | $34.49_{\pm 0.2}$ | $40.27_{\pm 0.2}$ |
| *Federated TTA Methods* | | | | |
| ATP | $41.41_{\pm 0.2}$ | $50.29_{\pm 0.2}$ | $37.25_{\pm 0.2}$ | $\underline{42.98}_{\pm 0.2}$ |
| FedSPL | $37.52_{\pm 0.3}$ | $48.99_{\pm 0.1}$ | $34.36_{\pm 0.2}$ | $40.29_{\pm 0.2}$ |
| FedCTTA | $37.92_{\pm 0.2}$ | $43.86_{\pm 0.2}$ | $37.20_{\pm 0.2}$ | $39.66_{\pm 0.2}$ |
| **FedPAT (Ours)** | $\mathbf{42.55}_{\pm 0.2}$ | $\mathbf{52.12}_{\pm 0.2}$ | $\mathbf{41.05}_{\pm 0.2}$ | $\mathbf{45.24}_{\pm 0.2}$ |

*Table 3.* Test-time adaptation accuracy (%) on CIFAR-100-C with severity = 3 corruption severity across various distribution shifts.

| Method | FS | LS | HS | Avg. |
|---|---|---|---|---|
| *General TTA Methods* | | | | |
| No Adapt | $34.15_{\pm 0.3}$ | $51.44_{\pm 0.2}$ | $31.12_{\pm 0.3}$ | $38.24_{\pm 0.2}$ |
| BN-Adapt | $41.29_{\pm 0.2}$ | $55.44_{\pm 0.2}$ | $38.45_{\pm 0.3}$ | $45.06_{\pm 0.2}$ |
| SHOT | $26.42_{\pm 0.4}$ | $44.99_{\pm 0.3}$ | $24.92_{\pm 0.4}$ | $32.11_{\pm 0.3}$ |
| Tent | $40.35_{\pm 0.2}$ | $48.00_{\pm 0.2}$ | $37.19_{\pm 0.2}$ | $41.85_{\pm 0.2}$ |
| T3A | $41.28_{\pm 0.2}$ | $55.45_{\pm 0.2}$ | $38.49_{\pm 0.3}$ | $45.07_{\pm 0.2}$ |
| MEMO | $\underline{42.58}_{\pm 0.2}$ | $55.21_{\pm 0.2}$ | $39.53_{\pm 0.2}$ | $45.77_{\pm 0.2}$ |
| EM | $39.00_{\pm 0.3}$ | $54.34_{\pm 0.2}$ | $37.95_{\pm 0.3}$ | $43.76_{\pm 0.2}$ |
| BBSE | $12.85_{\pm 0.4}$ | $52.14_{\pm 0.3}$ | $13.09_{\pm 0.4}$ | $26.03_{\pm 0.3}$ |
| Surgical | $40.15_{\pm 0.2}$ | $\underline{55.78}_{\pm 0.2}$ | $37.99_{\pm 0.3}$ | $44.64_{\pm 0.2}$ |
| *Federated TTA Methods* | | | | |
| ATP | $42.33_{\pm 0.2}$ | $55.45_{\pm 0.2}$ | $\underline{39.98}_{\pm 0.2}$ | $\underline{45.92}_{\pm 0.2}$ |
| FedSPL | $41.06_{\pm 0.2}$ | $55.55_{\pm 0.2}$ | $32.36_{\pm 0.3}$ | $42.99_{\pm 0.2}$ |
| FedCTTA | $39.52_{\pm 0.3}$ | $48.99_{\pm 0.2}$ | $37.33_{\pm 0.2}$ | $41.95_{\pm 0.2}$ |
| **FedPAT** | $\mathbf{43.54}_{\pm 0.2}$ | $\mathbf{55.88}_{\pm 0.2}$ | $\mathbf{40.08}_{\pm 0.2}$ | $\mathbf{46.50}_{\pm 0.2}$ |

ATP baseline method, our method consistently achieves improvements across all three benchmark datasets. Although the gains appear slight, they are statistically meaningful under such challenging conditions. Compared to the recently proposed FedSPL and FedCTTA methods that explicitly address the shift problem, FedPAT achieves outstanding results across various scenarios. On Tiny-ImageNet-C, which contains a larger number of classes, FedPAT yields an average improvement of approximately 4% over different shift types. FedPAT consistently surpasses traditional TTA methods while being more computationally efficient than MEMO that requires multiple augmented forward passes per sample.

We additionally assess the robustness of FedPAT under randomly sampled corruption severities on unseen clients using the Tiny-ImageNet-C dataset across various shift settings, as reported in Table 2. FedPAT consistently outperforms the ATP baseline, achieving an average improvement of approximately 3%. In contrast, existing federated TTA methods such as FedCTTA and FedSPL may face increasing chal-

lenges in large-class scenarios, as the aggregation of fitted noise distributions or contrastive representations becomes less reliable. Among general TTA approaches, MEMO remains competitive when integrated into the FL framework.

Table 3 presents test-time adaptation results on CIFAR-100-C with corruption severity set to 3. The observed trends are consistent with those under higher corruption levels, while FedPAT exhibits stronger relative gains when the corruption is less severe. This suggests that the global prototype affinity topology can be estimated more reliably on cleaner data, thereby enabling more effective topology-guided adaptation across different distribution shifts.

### 5.3. Visualization of Global PAT (RQ2)

**Construction of Global PAT.** To understand the consistency knowledge learned by FedPAT, we visualize how class-wise aggregation consolidates heterogeneous local prototypes from 240 source clients into consensus global prototypes,

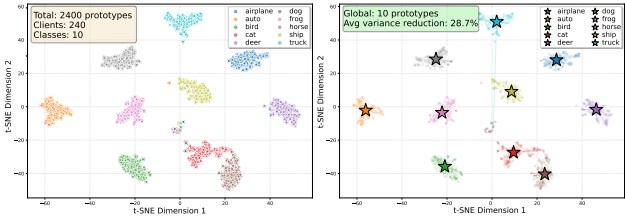

**Figure 3. Left:** Local prototypes from 240 source clients exhibit scatter due to client-specific feature spaces. **Right:** Global prototypes (★) consolidate local prototypes via sample-weighted averaging, achieving 28.7% variance reduction.

in conjunction with the learned implicit PAT. Figure 3 illustrates the prototype aggregation process on the CIFAR-10-C dataset under feature distribution shifts, visualized using t-SNE embeddings. Before aggregation (left), 2,400 local prototypes from 240 clients show substantial intra-class dispersion due to heterogeneous corruptions, despite maintaining clear inter-class boundaries. After aggregation (right), the weighted average merges these prototypes into 10 global prototypes, achieving a 28.7% reduction in intra-class variance while maintaining class separability. This indicates that weighted aggregation effectively captures consensus representations across heterogeneous clients.

**Global PAT Reveals Meaningful Structure.** We visualize the learned global PAT $\mathbf{A}_G$ on the CIFAR-10-C dataset. Figure 4a presents the $10 \times 10$ affinity matrix in the form of a heatmap. The matrix exhibits a clear block-diagonal structure, revealing meaningful semantic similarities between classes. For instance, the *automobile–truck* pair forms the strongest vehicle-related cluster with an affinity score of 0.93, reflecting their shared ground-vehicle characteristics and highly similar visual appearances. Similarly, animal classes demonstrate particularly high mutual affinities, with the *cat–dog* pair achieving an affinity of 0.97, which is consistent with their shared mammalian traits and similar fur textures. Interestingly, a relatively high affinity (0.87) is also observed between the *airplane* and *bird* classes. We hypothesize that this arises from shared contextual and visual cues, such as sky-dominated backgrounds and flight-related features learned during training.

Figure 4b formalizes the PAT as an undirected graph, where nodes represent class prototypes and edges connect semantically affine classes. These visualizations indicate that the global PAT captures genuine structures aligned with human-defined taxonomies, rather than spurious correlations induced by the federated optimization process.

**Cross-Client Stability of Prototype Affinity Topology.** To empirically validate that PAT captures cross-client invariant structural knowledge, we compare inter-client consistency across three complementary metrics: topological consistency via Spearman rank correlation (Soundarajan et al., 2014) between prototype affinity matrices, feature-level con-

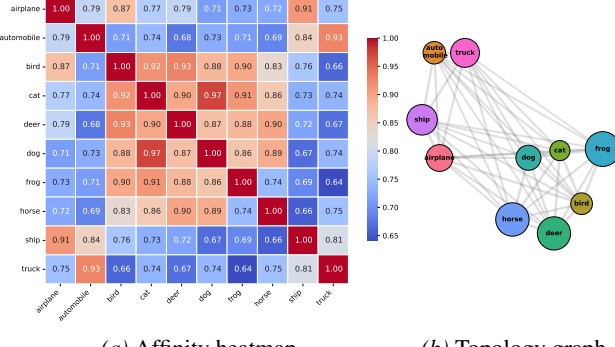

*(a) Affinity heatmap*   *(b) Topology graph*

**Figure 4.** Visualization of global PAT on CIFAR-10. **(a)** Heatmap of affinity matrix with color intensity indicating inter-class affinity strength (darker red = higher affinity). **(b)** Graph representation with nodes as class prototypes and edges encoding affinities.

sistency via Maximum Mean Discrepancy (MMD) (Long et al., 2015), and prediction-level consistency via symmetric KL divergence (Kou et al., 2024; 2025b;a), with the latter two normalized into comparable similarity scores.

As reported in Table 4, on CIFAR-10-C under feature shift, feature distributions exhibit substantial inter-client heterogeneity (mean = 0.34, std = 0.09), largely due to incomplete class support at local sites. In contrast, PAT maintains remarkably high consistency (mean = 0.87, std = 0.03), achieving $2.60\times$ and $1.55\times$ improvements over feature-level and prediction-level consistency, respectively. Under the more challenging hybrid shift on CIFAR-100-C with 100 classes, PAT consistency remains competitive at 0.69 (std = 0.16), still outperforming feature consistency by $2.12\times$. The heatmaps in Figure 5 further confirm that PAT maintains uniformly high inter-client affinity while feature-level similarities exhibit pronounced spatial divergence. These results demonstrate that class relational structure provides a more stable cross-client anchor than local feature statistics or model outputs, supporting its role as a robust global structural prior. This stability also holds under extreme domain gaps (DomainNet Real→Quickdraw, +5.31% over No Adapt; see Appendix B.2.4), with theoretical justification provided in Appendix C.1.

### 5.4. Ablation Study (RQ3)

**Component Contribution Analysis.** To validate the contribution of each component in FedPAT, we perform ablation studies on CIFAR-10-C with severity = 5 corruptions across various shifts, as listed in Table 5. As FedPAT fundamentally relies on the PAT design, the variant without topology modeling (w/o PAT) essentially reduces to *BN-Adapt* method. The performance degradation is particularly pronounced under *LS*, indicating that PAT effectively addresses class imbalance through its prototype-level relational encoding. In various distribution shift scenarios, the two components

*Table 4.* Inter-client consistency comparison across different metrics under moderate and severe distribution shifts. PAT maintains consistently higher inter-client consistency than feature-level and prediction-level statistics, especially under severe shifts.

*(a)* CIFAR-10-C (Feature Shift, Severity = 5)

| Metric | Mean | Min | Max | Std |
|---|---|---|---|---|
| PAT (Ours) | **0.875** | 0.847 | 0.917 | 0.026 |
| Feature Distribution | 0.336 | 0.206 | 0.491 | 0.090 |
| Prediction Consistency | 0.565 | 0.310 | 0.818 | 0.163 |
| *Relative Improvement (Mean)* | | | | |
| vs. Feature Dist. | **2.60×** | – | – | – |
| vs. Prediction | **1.55×** | – | – | – |

*(b)* CIFAR-100-C (Hybrid Shift, Severity = 5)

| Metric | Mean | Min | Max | Std |
|---|---|---|---|---|
| PAT (Ours) | **0.698** | 0.300 | 0.785 | 0.164 |
| Feature Distribution | 0.329 | 0.072 | 0.459 | 0.222 |
| Prediction Consistency | 0.433 | 0.202 | 0.543 | 0.078 |
| *Relative Improvement (Mean)* | | | | |
| vs. Feature Dist. | **2.12×** | – | – | – |
| vs. Prediction | **1.61×** | – | – | – |

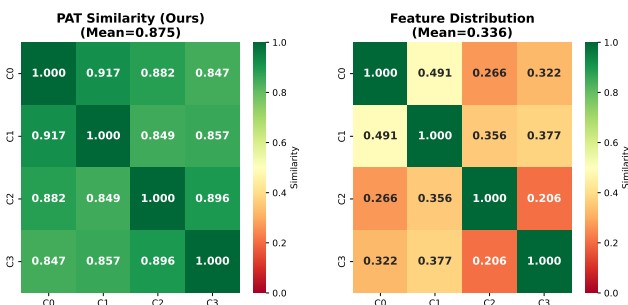 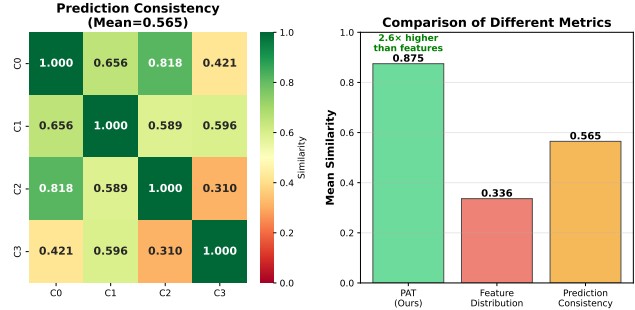

*Figure 5.* Inter-client consistency comparison. Despite significant heterogeneity in the feature distribution across clients (mean = 0.336), PAT's consistency is still 2.60× higher than the feature distribution (mean = 0.875), revealing its cross-client invariant structural knowledge. Three heatmaps show the pairwise similarity matrices for four randomly selected clients, while a bar chart summarizes the average similarity across all metrics.

*Table 5.* Ablation study on CIFAR-10-C (severity = 5).

| | *FS* | *LS* | *HS* | Avg. |
|---|---|---|---|---|
| **FedPAT** | $67.83_{\pm 0.1}$ | $80.73_{\pm 0.1}$ | $63.13_{\pm 0.2}$ | $70.56_{\pm 0.1}$ |
| w/o PAT | $63.48_{\pm 0.1}$ | $72.66_{\pm 0.2}$ | $56.53_{\pm 0.2}$ | $64.22_{\pm 0.2}$ |
| *Test-time Components* | | | | |
| w/o $p_{\text{TNI}}$ | $66.68_{\pm 0.2}$ | $78.38_{\pm 0.3}$ | $61.29_{\pm 0.1}$ | $68.78_{\pm 0.2}$ |
| w/o $p_{\text{cls}}$ | $65.58_{\pm 0.1}$ | $78.36_{\pm 0.1}$ | $61.73_{\pm 0.2}$ | $68.56_{\pm 0.1}$ |
| w/o Validation | $66.98_{\pm 0.1}$ | $80.22_{\pm 0.2}$ | $62.01_{\pm 0.1}$ | $69.73_{\pm 0.1}$ |

provide complementary gains, contributing different performance improvements. Notably, even when using PAT constructed solely from prototypes learned during pre-training (w/o Validation), the model still achieves stable performance improvements. This validates that federated pre-training has learned cross-client consistent topological structure, which can be directly used as global priors for test-time adaptation in the absence of validation data. Comprehensive sensitivity analyses on hyperparameters and validation round counts are detailed in Appendix B.2.7.

## 6. Conclusions

In this paper, we present FedPAT, a federated test-time adaptation framework that leverages prototype affinity topology as a global structural prior. FedPAT integrates topology-aware nonparametric inference with parametric classifier

to enable effective adaptation to unseen client distributions. Empirical results on three corrupted benchmarks demonstrate consistent improvements over representative federated and traditional TTA baselines. These findings suggest that explicitly modeling cross-client class relationships provides a promising direction for robust adaptation in open federated learning systems.

## Acknowledgements

This research was supported by the Jiangsu Science Foundation (BG2024036, BK20243012, BK20241297), the National Science Foundation of China (62406066, 62125602, U24A20324, 92464301), the New Cornerstone Science Foundation through the XPLORER PRIZE, and the Fundamental Research Funds for the Central Universities (2242025K30024), and the Big Data Computing Center of Southeast University.

## Impact Statement

This paper presents work whose goal is to advance the field of Machine Learning. There are many potential societal consequences of our work, none which we feel must be specifically highlighted here.

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

# Supplement for **FedPAT: Federated Test-Time Adaptation via Prototype Affinity Topology**

This appendix provides supplementary materials organized as follows:

- **Appendix A: Algorithm Details**
  Complete algorithmic description of FedPAT, including pseudocode, complexity analysis, and privacy Analysis.

- **Appendix B: Experimental Supplement**
  Additional experimental results, including robustness analysis under random corruption severity, effectiveness across different model architectures, convergence analysis, and comprehensive ablation studies.

- **Appendix C: Theoretical Analysis**
  Rigorous theoretical justification for cross-client stability of prototype affinity topology and graph-regularized interpretation of topology-aware diffusion.

## A. Algorithm Details

Algorithm 1 summarizes the complete FedPAT framework, integrating federated pre-training with PAT construction and test-time adaptation for unseen clients. Compared to existing federated learning approaches, FedPAT introduces only lightweight additional components, namely class prototypes and the prototype affinity topology. These components are transmitted to the target client only once during initialization, requiring no additional communication rounds.

Previous research on federated prototype learning (Tan et al., 2022a; Guo et al., 2024b) has shown that sharing class prototypes effectively preserves privacy because the prototype generalizes the distribution at the class level without revealing information about individual samples. Similarly, PAT captures aggregated inter-class relationship information only in the form of a topology matrix, without including any sample-specific data. Therefore, FedPAT follows the privacy assumptions commonly used in prototype-based federated learning and federated TTA methods, while introducing very little additional information compared to existing methods.

## B. Experimental Supplement

### B.1. Experimental Setup

All experiments are conducted in PyTorch using stochastic gradient descent. During federated pre-training, the learning rate is set to 0.1 with a batch size of 20. For the test-time adaptation phase, only the batch normalization (BN) layer parameters and classifier are updated, and the learning rate is set to 0.001. All methods undergo federated pre-training with 240 source clients over 200 communication epochs. Following ATP, we implement all general TTA baselines within the federated learning framework. The meta-learning stage of ATP is trained for 50 epochs, while the validation refinement process in our method uses only 5 epochs. During test-time adaptation, we set $k = 3$ for $k$NN and fix $\lambda_{\text{TCA}} = \lambda_{\text{ICC}} = 0.5$, and set $\tau_{\text{cls}} = 0.5$, with additional sensitivity analyses provided.

**Distribution Shift Scenarios.** To comprehensively evaluate robustness, we simulate three types of distribution shifts:

- **Feature Shift (*FS*).** Following (Bao et al., 2023), we randomly apply 15 different corruption types to source clients during pre-training, then evaluate on 4 *unseen* corruption types (Speckle noise, Gaussian blur, Spatter, Saturate) at severity level 5 for target clients, as illustrated in Figure 6. Each client possesses a distinct subset of classes rather than having access to the complete label space. This setting tests whether the learned global PAT generalizes to novel corruption types not encountered during training.

- **Label Shift (*LS*).** We employ a stepwise partitioning approach to introduce heterogeneity in label distribution among clients. Each client's class distribution follows a highly imbalanced long-tail distribution. For CIFAR-10-C, each

---

**Algorithm 1** FedPAT: Federated Test-Time Adaptation via Prototype Affinity Topology

---

1: **Input:** Source clients $\{S_i\}_{i=1}^N$ with $\{\mathcal{D}_{S_i}, \mathcal{V}_{S_i}\}$; target client $U_j$ with unlabeled data $\mathcal{X}_{U_j}$
2: **Output:** Predictions on $\mathcal{X}_{U_j}$
3: **/\* PAT Construction\*/**
4: Obtain pre-trained global model $\mathbf{w}_G$ via standard federated learning
5: **for** each source client $S_i \in \{S_1, \dots, S_N\}$ **do**
6:      Extract local prototypes $\{\mathbf{p}_c^{S_i}\}_{c=1}^C$ from $\mathcal{D}_{S_i}$ via Eq. (2)
7: **end for**
8: **Server:** Aggregate global prototypes $\{\mathbf{p}_c^G\}_{c=1}^C$ via Eq. (3)
9: **Server:** Construct global PAT $\mathbf{A}_G$ via Eq. (4)
10: **/\* Validation Refinement \*/**
11: **for** each source client $S_i$ **do**
12:      Adapt BN layers on $\mathcal{V}_{S_i}$ via TCA and ICC losses (Eq. (5))
13:      Upload adapted BN parameters $\{\gamma_i, \beta_i, \mu_i, \sigma_i^2\}$
14: **end for**
15: **Server:** Aggregate BN parameters via Eq. (9) $\rightarrow \mathbf{w}_G^*$
16: **/\* Target Test-Time Adaptation \*/**
17: Target client $U_j$ receives $\mathbf{w}_G^*$, $\{\mathbf{p}_c^G\}_{c=1}^C$, and $\mathbf{A}_G$
18: **for** each test batch $\mathcal{B} \subseteq \mathcal{X}_{U_j}$ **do**
19:      **for** each sample $\mathbf{x} \in \mathcal{B}$ **do**
20:          Topology diffusion: $p_{\text{TNI}}(y \mid \mathbf{x})$ by Eq. (11)
21:          Classifier prediction: $p_{\text{cls}}(y \mid \mathbf{x})$ by Eq. (12)
22:          Confidence-gated fusion: $p_{\text{final}}(y \mid \mathbf{x})$ by Eq. (14)
23:      **end for**
24:      Update BN and classifier parameters via gradient descent on $\mathcal{L}_{\text{EM}}$ by Eq. (15)
25: **end for**
26: **Return** Predictions $\{p_{\text{final}}(\cdot \mid \mathbf{x})\}_{\mathbf{x} \in \mathcal{X}_{U_j}}$

---

client contains 8 minor classes with 5 samples per class and 2 major classes with 80 samples per class. Similarly, for Tiny-ImageNet-C, each client has 40 major classes, with the sample ratio between major and minor classes being 16:1.

- **Hybrid Shift (*HS*).** We combine stepwise label partitioning with feature corruption: the source clients have imbalanced class distribution and corrupted features, while the target client faces data with the same imbalanced distribution and an unprecedented type of feature corruption. All clients contain samples spanning the complete label space, reflecting a highly challenging real-world deployment scenario. Each source client is allocated 160 labeled training samples and 40 labeled validation samples (200 samples in total), while each target client is provided with 200 unlabeled test samples.

## B.2. Additional Experiments

### B.2.1. ROBUSTNESS TO RANDOM CORRUPTION SEVERITY

We evaluate FedPAT against 12 competitive baseline methods on three benchmark datasets under randomly sampled corruption severities (levels 1–5). The results are summarized in Table 6. FedPAT achieves the best or near-best performance across most settings, with average accuracies of 75.63% on CIFAR-10-C and 45.41% on CIFAR-100-C. Compared to the strongest federated baseline ATP, FedPAT yields consistent improvements of 0.40% and 1.73%, respectively. Notably, larger gains are observed on more challenging datasets with finer-grained class distributions, highlighting the advantage of topology-aware adaptation in complex classification scenarios. Moreover, FedPAT consistently outperforms classical TTA methods, validating the importance of explicitly modeling inter-class semantic structures via the prototype affinity topology. FedPAT demonstrates robust performance across all types of distribution shifts. Under feature shift, the global prototypes aggregated from heterogeneous clients provide stable reference representations for adaptation. Under label shift, FedPAT achieves substantial improvements, indicating that topology-guided adaptation effectively constrains predictions to respect semantic relationships among classes. In the hybrid shift setting, FedPAT maintains competitive and stable performance, although slightly below ATP, which benefits from an extensive meta-learning process involving 50 training epochs, whereas FedPAT relies only on lightweight fine-tuning. All reported results are averaged over three random seeds.

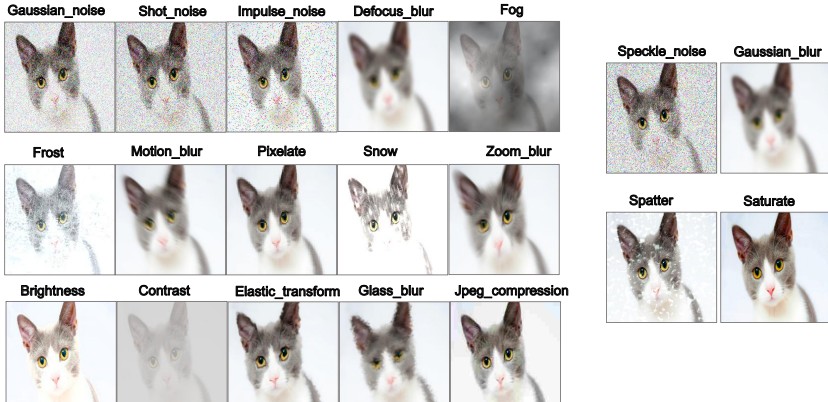

*Figure 6.* Visualization of corruption types. The left panel displays 15 known corruption types employed for source clients, whereas the right panel illustrates 4 unknown corruption types applied to unseen target clients.

*Table 6.* Test-time adaptation accuracy (%) under random corruption severity across three distribution shift types. Best in **bold**, second best underlined.

| Method | CIFAR-10-C | | | | CIFAR-100-C | | | |
|---|---|---|---|---|---|---|---|---|
| | *FS* | *LS* | *HS* | *Avg.* | *FS* | *LS* | *HS* | *Avg.* |
| *General TTA Baselines* | | | | | | | | |
| No Adapt | $70.57_{\pm 0.2}$ | $48.95_{\pm 0.2}$ | $64.06_{\pm 0.2}$ | $61.19_{\pm 0.2}$ | $41.22_{\pm 0.2}$ | $44.79_{\pm 0.2}$ | $39.74_{\pm 0.2}$ | $41.92_{\pm 0.2}$ |
| BN-Adapt | $72.91_{\pm 0.2}$ | $50.97_{\pm 0.2}$ | $47.36_{\pm 0.2}$ | $57.08_{\pm 0.2}$ | $43.16_{\pm 0.2}$ | $44.27_{\pm 0.2}$ | $20.71_{\pm 0.3}$ | $36.04_{\pm 0.2}$ |
| SHOT | $33.83_{\pm 0.3}$ | $30.53_{\pm 0.3}$ | $22.70_{\pm 0.3}$ | $29.02_{\pm 0.3}$ | $5.03_{\pm 0.3}$ | $6.11_{\pm 0.3}$ | $3.02_{\pm 0.3}$ | $4.72_{\pm 0.3}$ |
| Tent | $71.29_{\pm 0.2}$ | $49.91_{\pm 0.2}$ | $45.82_{\pm 0.2}$ | $55.67_{\pm 0.2}$ | $40.51_{\pm 0.2}$ | $41.66_{\pm 0.2}$ | $24.12_{\pm 0.3}$ | $35.43_{\pm 0.2}$ |
| T3A | $70.75_{\pm 0.2}$ | $71.57_{\pm 0.2}$ | $62.70_{\pm 0.2}$ | $68.34_{\pm 0.2}$ | $41.14_{\pm 0.2}$ | $48.78_{\pm 0.2}$ | $39.62_{\pm 0.2}$ | $43.18_{\pm 0.2}$ |
| MEMO | $73.53_{\pm 0.2}$ | $77.12_{\pm 0.2}$ | $68.62_{\pm 0.2}$ | $73.09_{\pm 0.2}$ | $\underline{43.57}_{\pm 0.2}$ | $\underline{50.88}_{\pm 0.2}$ | $\underline{40.82}_{\pm 0.2}$ | $\underline{45.15}_{\pm 0.2}$ |
| EM | $67.89_{\pm 0.2}$ | $\underline{80.71}_{\pm 0.2}$ | $70.03_{\pm 0.2}$ | $72.88_{\pm 0.2}$ | $38.96_{\pm 0.2}$ | $49.40_{\pm 0.2}$ | $39.40_{\pm 0.2}$ | $42.58_{\pm 0.2}$ |
| BBSE | $66.06_{\pm 0.2}$ | $78.96_{\pm 0.2}$ | $68.64_{\pm 0.2}$ | $71.22_{\pm 0.2}$ | $15.55_{\pm 0.3}$ | $23.30_{\pm 0.3}$ | $8.18_{\pm 0.3}$ | $15.68_{\pm 0.3}$ |
| Surgical | $69.33_{\pm 0.2}$ | $77.09_{\pm 0.2}$ | $67.05_{\pm 0.2}$ | $71.16_{\pm 0.2}$ | $40.14_{\pm 0.2}$ | $48.98_{\pm 0.2}$ | $39.65_{\pm 0.2}$ | $42.92_{\pm 0.2}$ |
| *Federated TTA Methods* | | | | | | | | |
| ATP | $\underline{73.83}_{\pm 0.2}$ | $79.95_{\pm 0.2}$ | $\mathbf{71.90}_{\pm 0.2}$ | $\underline{75.23}_{\pm 0.2}$ | $43.41_{\pm 0.2}$ | $48.83_{\pm 0.2}$ | $38.80_{\pm 0.2}$ | $43.68_{\pm 0.2}$ |
| FedSPL | $63.10_{\pm 0.3}$ | $74.24_{\pm 0.2}$ | $57.90_{\pm 0.2}$ | $65.08_{\pm 0.2}$ | $39.88_{\pm 0.2}$ | $48.55_{\pm 0.2}$ | $33.37_{\pm 0.2}$ | $40.60_{\pm 0.2}$ |
| FedCTTA | $69.37_{\pm 0.2}$ | $53.76_{\pm 0.2}$ | $49.09_{\pm 0.2}$ | $57.41_{\pm 0.2}$ | $39.63_{\pm 0.2}$ | $42.10_{\pm 0.2}$ | $38.98_{\pm 0.2}$ | $40.24_{\pm 0.2}$ |
| **FedPAT (Ours)** | $\mathbf{73.88}_{\pm 0.2}$ | $\mathbf{81.81}_{\pm 0.2}$ | $\underline{71.21}_{\pm 0.2}$ | $\mathbf{75.63}_{\pm 0.2}$ | $\mathbf{44.21}_{\pm 0.2}$ | $\mathbf{51.05}_{\pm 0.2}$ | $40.98_{\pm 0.2}$ | $\mathbf{45.41}_{\pm 0.2}$ |

### B.2.2. ADDITIONAL EXPERIMENT ON MISSING CLASSES AND DATA IMBALANCE

To further evaluate robustness under more realistic heterogeneous settings, we conduct additional experiments on CIFAR-10 considering two common challenges in federated learning: (1) **missing categories per client**, where each client observes only a subset of classes, and (2) **imbalanced local data volumes**, where the number of samples varies across clients.

We simulate a heterogeneous environment with 240 clients. Each client contains samples from a subset of classes, leading to incomplete label coverage and moderately imbalanced class distributions across clients. Meanwhile, the number of samples per client varies significantly, introducing additional heterogeneity in local data availability. The resulting data statistics are illustrated in Figure 7, which visualizes the number of samples per client and the number of classes observed by each client.

We evaluate all methods under the same setting. As shown in Table 7, FedPAT achieves the best performance, demonstrating strong robustness to realistic federated heterogeneity. Despite incomplete label coverage and moderate imbalance in client data distributions, FedPAT consistently maintains superior adaptation capability. This setting better reflects practical federated scenarios where both label distribution skew and data volume variability coexist, further validating the effectiveness and robustness of our approach.

### B.2.3. EFFECTIVENESS ACROSS DIFFERENT MODEL ARCHITECTURES

To evaluate the robustness and generalizability of FedPAT across different model architectures, we conduct experiments using a ResNet50 backbone on CIFAR-100-C and a Vision Transformer (ViT-B/16) (Shi et al., 2026c; 2024; Liu et al., 2026a;b; Lin et al.; 2024) backbone on Tiny-ImageNet-C, both with corruption severity 5. For Vision Transformers, we

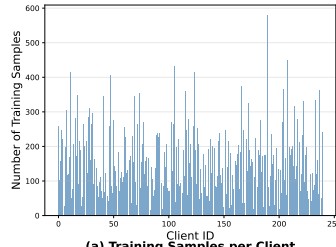
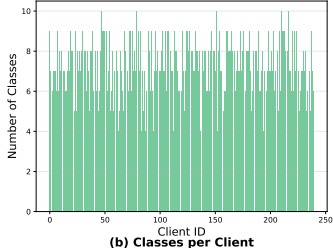

*Figure 7.* Client data distribution in the heterogeneous CIFAR-10 setting. (a) Training samples per client. (b) Classes per client.

*Table 7.* Test-time adaptation accuracy (%) on CIFAR-10 with missing categories and imbalanced client data.

| *General TTA* | | *Federated TTA* | |
| --- | --- | --- | --- |
| **Method** | **Acc.** | **Method** | **Acc.** |
| BN-Adapt | 65.83 | ATP | 68.35 |
| SHOT | 33.39 | FedSPL | 68.33 |
| T3A | 65.01 | FedCTTA | 56.98 |
| EM | 68.63 | | |
| BBSE | 68.65 | | |
| Surgical | 67.84 | | |
| **FedPAT (Ours): 70.28** | | | |

replace Batch Normalization with Layer Normalization (Ba et al., 2016) following the standard ViT architecture (Dosovitskiy et al., 2021; Niu et al., 2023; Shi et al., 2026a;b; Lin et al., 2026; 2023), and adapt the affine parameters of LayerNorm using the same optimization strategy. The results, averaged across three distribution shift scenarios (feature shift, label shift, and hybrid shift), are presented in Figure 8.

As shown in Figure 8(a), FedPAT achieves the highest average accuracy of 45.24% on CIFAR-100-C, substantially outperforming all baseline methods. Compared to the strongest federated TTA baseline ATP, FedPAT yields a 2.26 percentage-point improvement, demonstrating the effectiveness of topology-guided adaptation. FedPAT also surpasses the best-performing general TTA methods, including MEMO (42.56%) and EM (42.32%), validating that explicitly modeling inter-class semantic structure through PAT provides superior adaptation capabilities.

Figure 8(b) presents results on Tiny-ImageNet-C with a ViT-B/16 backbone, which poses a more challenging scenario due to the larger number of classes and the different architectural characteristics of transformers. FedPAT achieves the highest average accuracy of 54.94%, outperforming ATP by 1.87 percentage points. This improvement is particularly noteworthy given that several strong general TTA methods, such as FedSPL and Surgical, achieve competitive performance in this setting. The consistent superiority of FedPAT across both CNN and Transformer architectures demonstrates that the global prototype affinity topology provides a robust and architecture-agnostic structural prior for test-time adaptation.

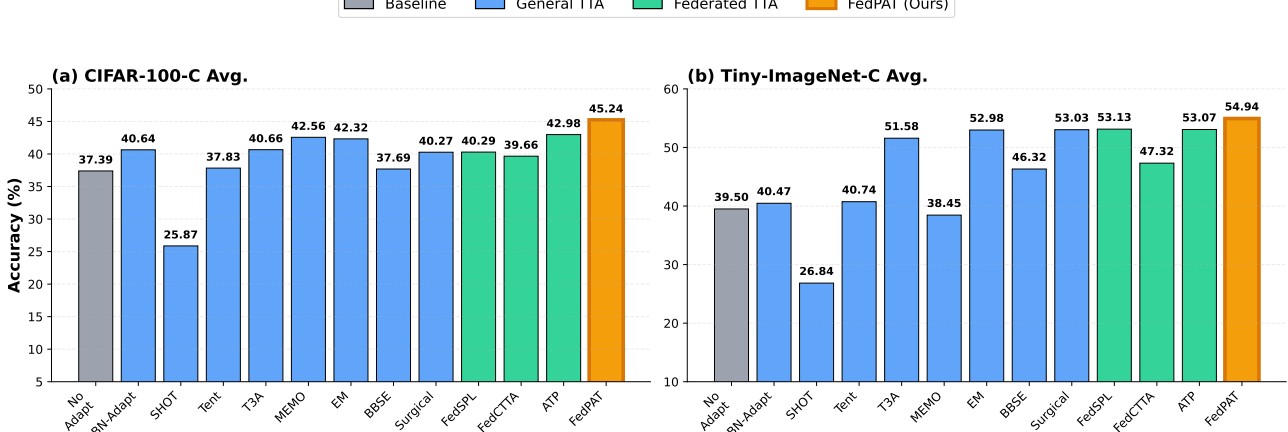

*Figure 8.* (a) Results on CIFAR-100-C with corruption severity = 5 using a ResNet50 backbone. (b) Results on Tiny-ImageNet-C with corruption severity = 5 using a Vision Transformer backbone.

### B.2.4. PAT INVARIANCE UNDER EXTREME SEMANTIC SHIFTS

To address potential concerns that PAT's effectiveness is an artifact of mild distribution shifts, we conducted a targeted experiment on **DomainNet** (Real → Quickdraw), one of the most severe semantic shift benchmarks, spanning 345 categories with 100 federated clients (80 source clients on Real, 20 target clients on Quickdraw). Even under this extreme domain gap, FedPAT achieves 62.96% accuracy compared to 57.65% for No Adapt, yielding a consistent improvement of +5.31%.

Although a *Quickdraw Cat* lacks fur and fine-grained texture, deep feature extractors still preserve a coarse relational

skeleton: its embedding remains geometrically closer to other animals than to vehicles or furniture in the representation space. This confirms that PAT captures **higher-level semantic proximity** beyond pixel-level visual similarity, and its effectiveness is not an artifact of weak distribution shifts.

### B.2.5. CONVERGENCE ANALYSIS

**Effective Convergence.** Both feature shift (Figure 9a) and hybrid shift (Figure 9b) settings demonstrate stable and consistent convergence within 200 communication rounds. The global model achieves approximately 50% accuracy on source clients, indicating that federated collaborative learning effectively aggregates knowledge from heterogeneous clients. This well-converged global model serves as a strong initialization for downstream adaptation, as it has learned generalizable feature representations and meaningful class prototypes from diverse source distributions.

**Performance Gap under Distribution Shift.** As illustrated by the green box in Figure 9, when the converged global model is directly deployed to unseen target clients without any adaptation, the performance drops significantly by approximately 20%, providing empirical validation for the description in Figure 1(a). This gap between source and target performance clearly demonstrates the distribution shift challenge: despite effective convergence on source clients, the model fails to generalize to target clients with different data distributions.

These observations lead to two important conclusions: the converged global model provides a reliable foundation with well-learned representations and topology structure, and additional adaptation is necessary to bridge the distribution gap at test time. Our proposed FedPAT framework leverages both aspects by using the pre-trained global model and the learned PAT as anchors, while performing efficient test-time adaptation to handle distribution shifts without requiring labeled target data or additional communication rounds.

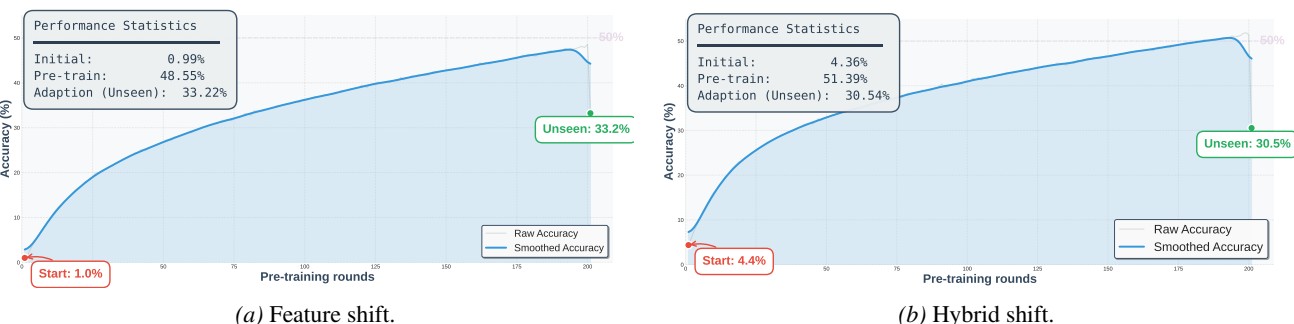

*(a)* Feature shift.        *(b)* Hybrid shift.

*Figure 9.* Convergence curves of federated pre-training on CIFAR-100-C under different distribution shifts. Both settings demonstrate stable convergence within 200 communication rounds.

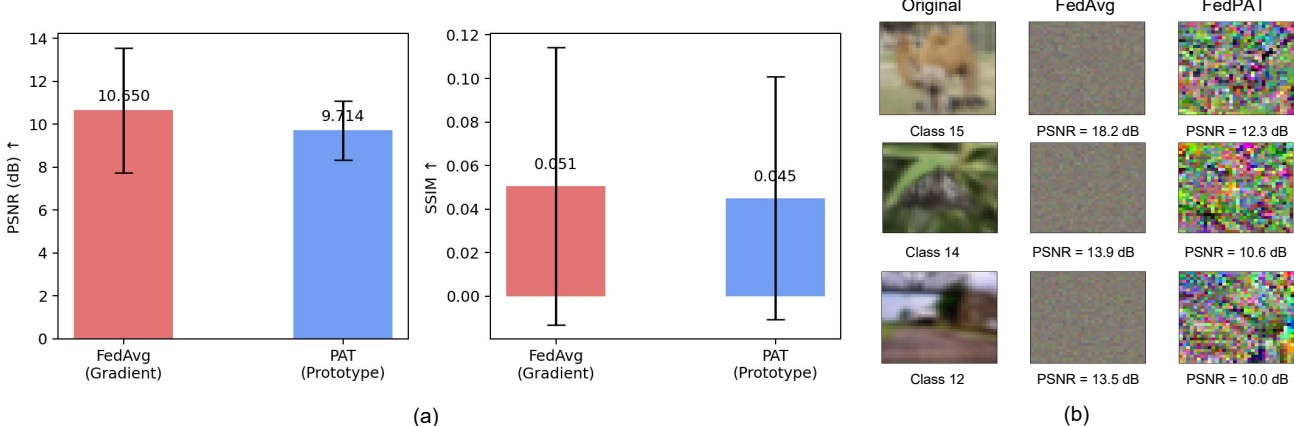

*Figure 10.* Privacy leakage comparison under model inversion attacks using DLG (Zhu et al., 2019) on CIFAR-100. (a) Average PSNR and SSIM over 500 attack rounds, computed across 50 samples. (b) Representative reconstructed images from FedAvg and PAT. Lower PSNR/SSIM indicates reduced reconstruction fidelity and improved privacy protection.

### B.2.6. PRIVACY ANALYSIS

To evaluate the privacy properties of PAT, we conduct a reconstruction-based privacy assessment and compare it with the standard FedAvg protocol. We perform 500 rounds of model inversion attacks using Deep Leakage from Gradients (DLG) on the CIFAR-100 dataset, and report the reconstruction fidelity averaged over 50 samples. Following common practice, we adopt Peak Signal-to-Noise Ratio (PSNR) and Structural Similarity Index (SSIM) as quantitative metrics, where higher values indicate more accurate reconstruction and therefore greater privacy leakage risk.

As illustrated in Figure 10, PAT consistently yields lower reconstruction fidelity than FedAvg across multiple representative classes. Since higher PSNR and SSIM indicate more accurate reconstruction and therefore greater privacy leakage risk, the reduced values achieved by PAT suggest improved privacy protection. For example, for Class 14, PSNR decreases from 13.9 dB (FedAvg) to 10.6 dB (PAT); for Class 12, PSNR decreases from 13.5 dB to 10.0 dB. Averaged across samples, PAT reduces PSNR from 10.55 dB to 9.71 dB and SSIM from 0.051 to 0.045. These consistently lower reconstruction metrics indicate that PAT exposes less recoverable information than FedAvg, demonstrating stronger resistance to model inversion attacks.

### B.2.7. ABLATION STUDIES

We conduct sensitivity analysis on the following hyperparameters: the number of neighbors $k$ in $k$NN-based non-parametric inference, the diffusion strength $\alpha$ in topology-aware diffusion, the validation loss weights $\lambda_{\text{TCA}}$ and $\lambda_{\text{ICC}}$, and the number of validation refinement epochs.

*Table 8.* Hyperparameter sensitivity analysis on CIFAR-100-C (%).

*(a) $k$NN neighbors $k$ (with $\alpha = 0.4$).*

| $k$ | Feat. | Label | Hybrid | Avg. |
|---|---|---|---|---|
| 1 | 38.35 | 49.16 | 33.98 | 40.50 |
| **3** | **38.99** | **50.31** | **34.62** | **41.31** |
| 5 | 38.23 | 49.06 | 33.92 | 40.40 |

*(b) Diffusion strength $\alpha$ (with $k = 3$).*

| $\alpha$ | Feat. | Label | Hybrid | Avg. |
|---|---|---|---|---|
| 0 | 37.84 | 48.26 | 33.67 | 39.92 |
| **0.4** | **38.99** | **50.31** | **34.62** | **41.31** |
| 1 | 38.19 | 48.22 | 33.76 | 40.06 |

**Effect of $k$ in $k$NN Inference.** As shown in Table 8a, FedPAT is robust to the choice of $k$, with performance variations less than 1% across different values. When $k = 1$, the prediction relies solely on the nearest prototype, making it slightly susceptible to prototype estimation noise. Increasing $k$ to 3 provides a better balance between discrimination and robustness. However, larger values ($k = 5$) introduce distant prototypes that may dilute discriminative information. We adopt $k = 3$ as the default setting.

**Diffusion Strength $\alpha$ in TNI.** Table 8b reveals more pronounced sensitivity to $\alpha$. When $\alpha = 0$ (no topology diffusion), the model achieves the lowest accuracy across all shift types, validating the effectiveness of our proposed TNI mechanism. The optimal setting $\alpha = 0.4$ improves average accuracy by 1.4% over $\alpha = 0$. However, relying entirely on topology propagation when $\alpha = 1$ leads to performance degradation due to over-smoothing; excessive diffusion dilutes class discrimination information. This confirms that moderate diffusion optimally balances noise reduction and information preservation.

### B.2.8. ANALYSIS OF VALIDATION REFINEMENT

**Hyperparameter Sensitivity.** We first analyze the sensitivity of FedPAT to the validation refinement loss weights $\lambda_{\text{TCA}}$ and $\lambda_{\text{ICC}}$ on CIFAR-100-C under hybrid distribution shift (severity = 5). As shown in Figure 11a, both losses achieve their best accuracy at $\lambda = 0.5$, reaching 34.72%. When either $\lambda_{\text{TCA}}$ or $\lambda_{\text{ICC}}$ is set to zero, performance drops noticeably, indicating that both topology consistency alignment and intra-class compactness are necessary for effective refinement. Moreover, the performance degrades smoothly when deviating from the optimal value, suggesting that FedPAT is not overly sensitive to precise hyperparameter tuning. These results support the robustness of the proposed validation refinement objective.

**Fast Convergence of Validation Refinement.** Figure 11b illustrates the convergence behavior of the validation refinement stage on CIFAR-100-C. Starting from an initial accuracy of 34.67%, performance improves rapidly and reaches 34.72% within the first five refinement epochs, after which it quickly saturates. By epoch 20, the accuracy further increases to 34.78%, yielding a marginal gain of less than 0.1%. This rapid convergence indicates that the global prototype affinity topology provides a strong structural initialization, enabling effective refinement with only a few optimization steps.

In practice, we observe that using five refinement epochs already achieves over 90% of the total performance gain, offering a

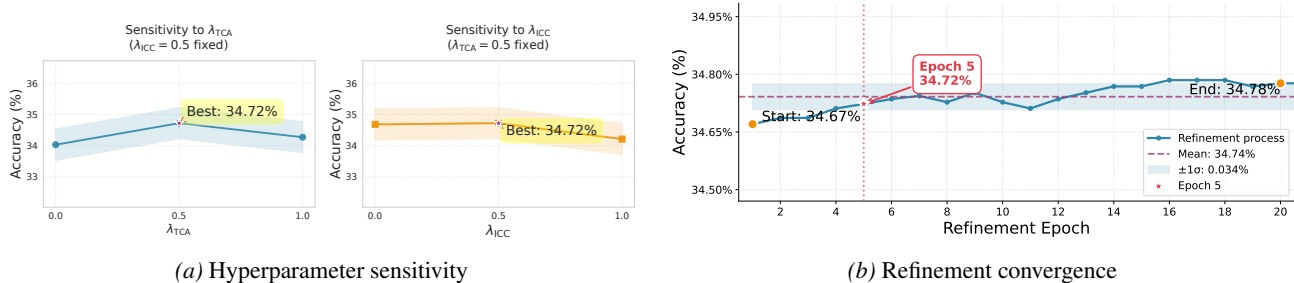

*(a)* Hyperparameter sensitivity        *(b)* Refinement convergence

*Figure 11.* Validation phase analysis. (a) Hyperparameter sensitivity on CIFAR-100-C hybrid shift: both $\lambda_{\text{TCA}}$ and $\lambda_{\text{ICC}}$ achieve optimal performance at 0.5. (b) Refinement convergence on CIFAR-100-C hybrid shift: accuracy stabilizes within 5 epochs, validating efficient adaptation without extensive fine-tuning.

favorable trade-off between accuracy and computational efficiency. Based on these observations, we set $\lambda_{\text{TCA}} = \lambda_{\text{ICC}} = 0.5$ and adopt five refinement epochs across all experiments. Moreover, in deployment scenarios where validation data is unavailable, this refinement stage can be omitted with only negligible impact on performance.

## C. Theoretical Analysis

This appendix provides a rigorous theoretical justification for FedPAT from two complementary perspectives: (i) the cross-client stability of Prototype Affinity Topology (PAT) under heterogeneous data distributions, and (ii) the graph-regularization interpretation of topology-aware diffusion used in test-time adaptation.

**Notation Clarification.** In the main paper, topology-aware diffusion is implemented using a row-normalized affinity matrix derived from the global PAT. In practice, this is realized by first enforcing non-negativity, adding self-loops, and then applying row normalization, resulting in a row-stochastic transition matrix. This formulation corresponds to a standard random-walk operator and induces first-order graph smoothing behavior.

*Table 9.* Summary of notation used in theoretical analysis.

| Symbol | Description |
|---|---|
| $C$ | Number of classes |
| $N$ | Number of source clients |
| $\mathbf{p}_c^i \in \mathbb{R}^d$ | $c$ class prototype for client $i$ (unit norm) |
| $\mathbf{p}_c^G \in \mathbb{R}^d$ | Global $c$ class prototype (unit norm) |
| $\mathbf{A}_i \in \mathbb{R}^{C \times C}$ | Local prototype affinity topology (PAT) for client $i$ |
| $\mathbf{A}_G \in \mathbb{R}^{C \times C}$ | Global prototype affinity topology |
| $\mathbf{T}_G \in \mathbb{R}^{C \times C}$ | Row-stochastic transition matrix derived from $\mathbf{A}_G$ |
| $\mathbf{L}_{\text{rw}} = \mathbf{I} - \mathbf{T}_G$ | Random-walk graph Laplacian |
| $\epsilon_p$ | Bound on prototype deviation |
| $p_{\text{proto}}$ | Prototype-based posterior prediction |
| $p_{\text{TNI}}$ | Topology-refined posterior via diffusion |
| $\alpha \in (0, 1)$ | Diffusion strength parameter |

### C.1. Cross-Client Stability of Prototype Affinity Topology

**Motivation.** Although feature representations may vary significantly across clients under distribution shifts, we empirically observe that inter-class relationships, captured by prototype affinity topology (PAT), remain highly consistent. This section formalizes and proves this observation.

#### C.1.1. AFFINITY CONSTRUCTION

Given $\ell_2$-normalized class prototypes $\{\mathbf{p}_c^i\}_{c=1}^C$ for client $i$ (i.e., $\|\mathbf{p}_c^i\|_2 = 1$), the affinity between classes $c$ and $c'$ is defined using non-negative cosine similarity:

$$\mathbf{A}_i[c, c'] = \max\left(0, \mathbf{p}_c^{i\top} \mathbf{p}_{c'}^i\right). \tag{16}$$

This construction removes negative correlations while preserving semantic proximity. The global affinity matrix $\mathbf{A}_G$ is defined analogously using global prototypes $\{\mathbf{p}_c^G\}_{c=1}^C$.

### C.1.2. PROTOTYPE PERTURBATION ASSUMPTION

**Assumption C.1** (Bounded Prototype Deviation). For each class $c$ and client $i$, the local prototype deviates from the global prototype by at most $\epsilon_p$:
$$\|\mathbf{p}_c^i - \mathbf{p}_c^G\|_2 \leq \epsilon_p,$$
where $\epsilon_p \in (0, 1)$ is a small positive constant.

*Remark* C.2. This assumption is reasonable in practice for the following reasons: The feature encoder is trained using federated learning, which aggregates knowledge from different clients and generates a shared feature space. The prototype is computed from multiple samples of each class, thus being robust to noise from individual samples.

### C.1.3. AFFINITY STABILITY

**Lemma C.3** (Affinity Perturbation Bound). *Under Assumption C.1, for any classes $c, c'$:*
$$\left| \mathbf{A}_i[c, c'] - \mathbf{A}_G[c, c'] \right| \leq 2\epsilon_p + \epsilon_p^2.$$

*Proof.* Since both local and global prototypes are unit-norm, we have:
$$
\begin{aligned}
\left| \langle \mathbf{p}_c^i, \mathbf{p}_{c'}^i \rangle - \langle \mathbf{p}_c^G, \mathbf{p}_{c'}^G \rangle \right| &= \left| \langle \mathbf{p}_c^i, \mathbf{p}_{c'}^i \rangle - \langle \mathbf{p}_c^G, \mathbf{p}_{c'}^i \rangle + \langle \mathbf{p}_c^G, \mathbf{p}_{c'}^i \rangle - \langle \mathbf{p}_c^G, \mathbf{p}_{c'}^G \rangle \right| \\
&\leq \left| \langle \mathbf{p}_c^i - \mathbf{p}_c^G, \mathbf{p}_{c'}^i \rangle \right| + \left| \langle \mathbf{p}_c^G, \mathbf{p}_{c'}^i - \mathbf{p}_{c'}^G \rangle \right| \\
&\leq \|\mathbf{p}_c^i - \mathbf{p}_c^G\|_2 \cdot \|\mathbf{p}_{c'}^i\|_2 + \|\mathbf{p}_c^G\|_2 \cdot \|\mathbf{p}_{c'}^i - \mathbf{p}_{c'}^G\|_2 \\
&\leq \epsilon_p \cdot 1 + 1 \cdot \epsilon_p = 2\epsilon_p.
\end{aligned}
\tag{17}
$$

**Correcting for non-unit norm.** Since $\|\mathbf{p}_c^i\|_2 = 1$ but $\|\mathbf{p}_c^i - \mathbf{p}_c^G\|_2 \leq \epsilon_p$, by the reverse triangle inequality:
$$\left| \|\mathbf{p}_c^i\|_2 - \|\mathbf{p}_c^G\|_2 \right| \leq \|\mathbf{p}_c^i - \mathbf{p}_c^G\|_2 \leq \epsilon_p.$$

Thus $\|\mathbf{p}_c^G\|_2 \geq 1 - \epsilon_p$. By normalizing, the effective inner product perturbation is bounded by:
$$\frac{2\epsilon_p}{(1 - \epsilon_p)^2} \approx 2\epsilon_p(1 + 2\epsilon_p) = 2\epsilon_p + 4\epsilon_p^2.$$

For small $\epsilon_p$, we use the looser but cleaner bound $2\epsilon_p + \epsilon_p^2$.

**Handling the max operator.** Let $a_i = \langle \mathbf{p}_c^i, \mathbf{p}_{c'}^i \rangle$ and $a_G = \langle \mathbf{p}_c^G, \mathbf{p}_{c'}^G \rangle$. Based on the steps above, we have $|a_i - a_G| \leq 2\epsilon_p + \epsilon_p^2$.

*Case 1:* If both $a_i \geq 0$ and $a_G \geq 0$, then
$$|\mathbf{A}_i[c, c'] - \mathbf{A}_G[c, c']| = |a_i - a_G| \leq 2\epsilon_p + \epsilon_p^2.$$

*Case 2:* If $a_i < 0$ and $a_G \geq 0$ (or vice versa), then one of $\mathbf{A}_i[c, c']$ or $\mathbf{A}_G[c, c']$ is zero. The perturbation is bounded by $\max(|a_i|, |a_G|) \leq 1$, but since $|a_i - a_G| \leq 2\epsilon_p + \epsilon_p^2$, we have $|\mathbf{A}_i[c, c'] - \mathbf{A}_G[c, c']| \leq 2\epsilon_p + \epsilon_p^2$.

*Case 3:* If both $a_i < 0$ and $a_G < 0$, then both affinities are zero, so the perturbation is zero. Combining all cases yields the desired bound. $\square$

## C.2. Graph-Regularized Interpretation of Topology-Aware Diffusion

We provide a principled interpretation of topology-aware non-parametric inference (TNI) through the lens of graph signal processing. Given $\ell_2$-normalized prototypes $\{\mathbf{p}_c^G\}_{c=1}^C$ (or $\{\mathbf{p}_c^i\}$ for client $i$), we define affinity using non-negative cosine similarity:
$$\mathbf{A}_G[c, c'] = \phi\big(\mathbf{p}_c^{G\top} \mathbf{p}_{c'}^G\big), \quad \phi(x) = \max(0, x). \tag{18}$$

This ReLU activation ensures $\mathbf{A}_G \geq 0$ by removing negative correlations, which can arise from orthogonal class pairs. To construct a row-stochastic transition matrix, we add self-loops and normalize:

$$\mathbf{T}_G = \mathbf{D}_G^{-1}\left(\mathbf{A}_G + \eta\mathbf{I}\right), \quad \mathbf{D}_G = \text{diag}((\mathbf{A}_G + \eta\mathbf{I})\mathbf{1}), \tag{19}$$

where $\eta > 0$ introduces self-loops to ensure well-defined random-walk dynamics and stabilize row normalization. In our implementation, we set $\eta = 1$, which corresponds to adding unit self-loops and empirically yields stable diffusion behavior. The same construction applies to each client, yielding $\mathbf{A}_i$ and $\mathbf{T}_i$.

### C.2.1. PRELIMINARIES: RANDOM-WALK LAPLACIAN

**Definition C.4** (Random-Walk Graph Laplacian). Given a row-stochastic transition matrix $\mathbf{T}_G$ constructed from the prototype affinity topology, the random-walk graph Laplacian is defined as

$$\mathbf{L}_{\text{rw}} = \mathbf{I} - \mathbf{T}_G.$$

*Remark* C.5. The random-walk Laplacian characterizes graph smoothness. For any signal $\mathbf{f} \in \mathbb{R}^C$, we have

$$\mathbf{f}^\top \mathbf{L}_{\text{rw}} \mathbf{f} = \sum_{c=1}^{C}\sum_{c'=1}^{C} \mathbf{T}_G[c, c'] \left(\mathbf{f}[c] - \mathbf{f}[c']\right)^2,$$

which measures the variation of $\mathbf{f}$ across edges in the prototype topology graph. A smaller value indicates that $\mathbf{f}$ varies smoothly among semantically related classes.

### C.2.2. GRAPH-REGULARIZED OBJECTIVE

Topology-aware diffusion admits a principled interpretation as a graph-regularized posterior refinement process. We consider the following optimization problem for refining prototype-based class posteriors:

$$\min_{\mathbf{p}\in\mathbb{R}^C} \underbrace{\|\mathbf{p} - p_{\text{proto}}\|_2^2}_{\text{fidelity term}} + \alpha \underbrace{\mathbf{p}^\top \mathbf{L}_{\text{rw}} \mathbf{p}}_{\text{graph smoothness term}}, \tag{20}$$

where $p_{\text{proto}}$ denotes the initial non-parametric prediction and $\alpha > 0$ is a diffusion coefficient governing the strength of topology regularization. The fidelity term preserves alignment with prototype similarity, while the smoothness term encourages semantically related classes to produce consistent posterior probabilities according to the global prototype affinity topology.

**Proposition C.6** (Closed-Form Solution). *The unique minimizer of* (20) *is given by*

$$\mathbf{p}^* = (\mathbf{I} + \alpha\mathbf{L}_{\text{rw}})^{-1} p_{\text{proto}}.$$

*Proof.* Taking the gradient of (20) with respect to $\mathbf{p}$ and setting it to zero yields

$$2(\mathbf{p} - p_{\text{proto}}) + 2\alpha\mathbf{L}_{\text{rw}}\mathbf{p} = 0,$$

which leads to $(\mathbf{I} + \alpha\mathbf{L}_{\text{rw}})\mathbf{p} = p_{\text{proto}}$. Since $\mathbf{L}_{\text{rw}}$ is positive semi-definite, the matrix $\mathbf{I} + \alpha\mathbf{L}_{\text{rw}}$ is strictly positive definite for $\alpha > 0$ and therefore invertible, ensuring a unique solution. $\square$

### C.2.3. CONNECTION TO TOPOLOGY-AWARE DIFFUSION

We show that the proposed topology-aware non-parametric inference (TNI) corresponds to a first-order approximation of the above graph-regularized solution.

**Theorem C.7** (Single-Step Diffusion as Graph Regularization). *The topology-aware diffusion step*

$$p_{\text{TNI}} = (1 - \alpha)\, p_{\text{proto}} + \alpha\, p_{\text{proto}} \mathbf{T}_G, \tag{21}$$

*is equivalent to*

$$p_{\text{TNI}} = (\mathbf{I} - \alpha\mathbf{L}_{\text{rw}})\, p_{\text{proto}}, \tag{22}$$

*which corresponds to the first-order Neumann-series approximation of* $(\mathbf{I} + \alpha\mathbf{L}_{\text{rw}})^{-1} p_{\text{proto}}$.

*Proof.* Starting from (21), we have

$$
\begin{aligned}
p_{\text{TNI}} &= (1 - \alpha) \, p_{\text{proto}} + \alpha \, p_{\text{proto}} \mathbf{T}_G \\
&= p_{\text{proto}} - \alpha p_{\text{proto}} (\mathbf{I} - \mathbf{T}_G) \\
&= (\mathbf{I} - \alpha \mathbf{L}_{\text{rw}}) \, p_{\text{proto}},
\end{aligned}
\tag{23}
$$

which establishes the equivalence. Moreover, applying the Neumann series expansion yields

$$
(\mathbf{I} + \alpha \mathbf{L}_{\text{rw}})^{-1} = \mathbf{I} - \alpha \mathbf{L}_{\text{rw}} + O(\alpha^2),
$$

showing that $p_{\text{TNI}}$ is a first-order approximation of the graph-regularized solution in Proposition C.6.  $\square$

### C.2.4. INTERPRETATION

**Corollary C.8** (Smoothness Effect). *Topology-aware diffusion reduces the graph smoothness energy:*

$$
p_{\text{TNI}}^{\top} \mathbf{L}_{\text{rw}} \, p_{\text{TNI}} \leq p_{\text{proto}}^{\top} \mathbf{L}_{\text{rw}} \, p_{\text{proto}}.
$$

*Remark* C.9 (Robustness Interpretation). Topology-aware diffusion acts as a low-pass filter on the prototype affinity graph: it suppresses high-frequency variations in class posteriors while preserving low-frequency semantic structure. As a result, TNI improves robustness under distribution shifts, since inter-class relationships encoded by the prototype affinity topology remain more stable than raw feature statistics.

