# OpenReview forum: "FedPAT: Federated Test-Time Adaptation via Prototype Affinity Topology"
_ICML.cc/2026/Conference — ICML 2026 regular_

### Official Review · Reviewer_LxYt · 2026-03-10

**Soundness:** 2
**Presentation:** 3
**Significance:** 2
**Originality:** 2
**Overall Recommendation:** 3
**Confidence:** 4

**Summary:**

This paper proposes FedPAT, a federated test-time adaptation framework for open-world federated learning. It observes that the prototype affinity topology (PAT)—the relational structure among class prototypes—remains stable across heterogeneous clients under distribution shifts. The method builds a global PAT by aggregating class prototypes from source clients on the server, uses topology-aware diffusion for non-parametric prediction during test time, fuses this prediction with the output of a parametric classifier, and performs lightweight optimization to adapt the model. Experiments are conducted on three corrupted image datasets (CIFAR-10-C, CIFAR-100-C, Tiny-ImageNet-C) to compare with recent federated TTA and classic TTA methods.

**Compliance With Llm Reviewing Policy:**

Affirmed.

**Final Justification:**

Thank you for the author's reply. However, I think this paper is a bit incremental. I decide to keep my score.

**Key Questions For Authors:**

- Could you provide the ablation results for predictions made solely using the prototype affinity diffusion (without fusing the classifier head)? If PAT is robust to OOD data, why is the classifier fusion necessary?
- How do you address the contradiction that if your feature extractor can produce a domain-invariant affinity matrix, it must already be extracting domain-invariant features, thereby contradicting the premise that the model lacks OOD generalization?
- Have you tested this method on datasets with severe semantic shifts (e.g., DomainNet: Real vs. Quickdraw)? Please provide an analysis of failure cases where the class affinity topology changes significantly between domains.
- Given that the PAT matrix provides structural priors to potential attackers, can you run a standard Model Inversion Attack (e.g., a variant of DLG) to quantify the actual privacy leakage (e.g., using MSE or PSNR of reconstructed images) compared to baseline FedAvg?

**Limitations:**

No. The authors do not adequately discuss critical limitations, including the unstable PAT in real OOD scenes, the ineffectiveness of pure PAT prediction, unaddressed privacy risks, and the large gap between experiments and real federated scenarios.

**Strengths And Weaknesses:**

## Strengths
- The paper targets a highly relevant and challenging scenario. Open-world FL with heterogeneous new clients is a realistic bottleneck for deploying FL systems.
- Relying on the relative topological structure (affinity between classes) rather than absolute feature coordinates is a conceptually appealing way to handle domain shifts.
- The overall mechanism (extracting prototypes -> building affinity matrix -> diffusion -> fusion -> lightweight update) is logically structured and easy to follow.

## Weaknesses
- There is a fundamental "chicken-and-egg" paradox in the paper's core motivation. The authors argue that the original model lacks Out-of-Distribution (OOD) generalization, hence the need for TTA. However, the proposed method heavily relies on the feature extractor to build a stable, invariant Prototype Affinity Topology (PAT). If the feature extractor is capable of maintaining a perfectly stable topological structure across different domains (i.e., extracting domain-invariant semantic features), then the feature extractor already possesses strong OOD generalization. In that case, a simple nearest-neighbor or linear probe would suffice. The paper fails to reconcile why a model with such a robust feature space needs complex affinity diffusion and TTA in the first place.

- The premise that class affinity remains invariant across domains is likely an artifact of weak, pixel-level domain shifts (e.g., Gaussian noise, blur). Let's take a simple 3-class example: Cat, Dog, and Human. In a dataset of real photos, Cats and Dogs are highly affine (furry, four legs). However, if the new domain consists of abstract "quickdraw" sketches, the visual affinity changes drastically. The observation of invariant affinity implies the datasets used might not have severe semantic shifts. A deeper analysis of failure cases where this topology collapses is missing.

- Prototypes and affinity matrices are standard tools. If the PAT-based prediction is truly robust to heterogeneous OOD data as claimed, it is unclear why the authors must fuse it with the predictions from the "fragile" classification head. This fusion often acts as a safety net when the proposed module (PAT) underperforms. And ablation study seems to confirms this.
Unsubstantiated Privacy Claims and Missing Threat Model: The paper heuristically claims that sharing prototypes and affinity matrices protects privacy. In the context of FL research in 2026, this is insufficient. It is well-documented that high-dimensional prototypes can be inverted to reveal sensitive attributes. This actually increases the attack surface for Model Inversion Attacks. The paper lacks a systematic threat model and empirical defense/attack evaluations.

---

> ### Author Rebuttal · Authors · 2026-03-31
>
> We thank the reviewer for the careful reading and valuable suggestions.
>
> ---
> **W3 & Q1: Ablation of PAT-only Prediction vs. Fusion**
>
> We thank the reviewer for this question. We apologize for the unclear presentation, which may have caused this misunderstanding. This ablation result already exists in Table 4, under the row labeled `w/o` $p_\text{cls}$, which removes the parametric classifier and relies solely on topology-aware diffusion.
> | Method | FS | LS | HS | Avg. |
> |-|-|-|-|-|
> | BN-Adapt | 63.48 | 72.66 | 56.53 | 64.22 |
> | w/o $p_\text{cls}$ (PAT only) | 65.58 | 78.36 | 61.73 | 68.56 |
> | FedPAT (full) | 67.83 | 80.73 | 63.13 | 70.56 |
>
> PAT-only inference outperforms BN-Adapt by a clear margin, confirming genuine robustness independent of the classifier. The fusion is **complementary by design**: TNI is robust but limited in fine-grained discriminability; the classifier is discriminative but shift-sensitive. The confidence gate $g$ dynamically selects the more reliable prediction per sample, providing additive rather than compensatory gains.
>
> ---
> **W1 & Q2: On the "Chicken-and-Egg" Paradox and Relational Invariance**
>
> We thank the reviewer for this profound conceptual challenge. We clarify that the perceived paradox arises from a **fundamental conflation** between two mathematically decoupled properties: _sample-level feature stability_ and _prototype-level relational stability_.
>
> 1. **Relational stability is a strictly weaker condition than feature invariance.** FedPAT does not assume that the feature extractor is fully domain-invariant. Under OOD shifts, the absolute coordinates of features in the latent space shift significantly, causing the pre-trained linear classifier to fail. The relative topology is inherently more stable than individual sample features. This distinction is directly supported by our empirical observation in **Table 5**: feature distribution similarity drops to **0.336**, whereas PAT similarity remains **0.875**, indicating a **2.60× relative stability advantage** at the relational level. This observation is consistent with prior work: Ni et al. [1] demonstrate that preserving inter-class topology significantly stabilizes continual TTA under distribution shift; You et al. [2] show that aligning second-order feature correlations effectively mitigates distribution shift.
>
> 2. **Empirical evidence confirms adaptation is necessary despite relational stability.** BN-Adapt, the closest proxy to a linear probe, achieves only **63.48%** under Feature Shift vs. FedPAT's **67.83%**. Furthermore, Figure 8 directly demonstrates that deploying the trained model without adaptation causes **~20% performance degradation**, empirically confirming that the model lacks OOD generalization despite maintaining stable relational structure. PAT exploits this relational stability not as a substitute for adaptation, but as a **structural prior that guides adaptation**.
>
> [1] Ni C et al. Maintaining consistent inter-class topology in continual test-time adaptation. CVPR, 2025. [2] You et al. Test-time Correlation Alignment. ICML, 2025.
>
> ---
>
> **W2 & Q3: PAT Invariance as Artifact of Weak Shifts**
>
> We thank the reviewer for the constructive challenge. To address this concern, we conducted a targeted experiment on DomainNet (Real to Quickdraw), one of the most severe semantic shifts, across 345 categories with 100 federated clients (**80 Source-Real, 20 Target-Quickdraw**).
> | Method | Test-time adaptation accuracy |
> |-|-|
> | No-Adapt | 57.65% |
> | **FedPAT (Ours)** | **62.96%** |
> | Improvement | **+5.31%** |
>
> Even under this extreme domain gap, FedPAT yields consistent improvements. Although a "Quickdraw Cat" lacks fur and texture, deep feature extractors still preserve a coarse relational skeleton, with its embedding remaining closer to other animals than to vehicles or furniture. This confirms that PAT captures higher-level semantic proximity beyond pixel-level visual similarity.
>
> ---
> **W4 & Q4: Privacy Claims and Model Inversion Attack**
>
> We acknowledge that our privacy analysis follows the heuristic verification protocol established in prior federated prototype learning works (Tan et al., 2022a; Guo et al., 2024). Based on the reviewer's suggestion, we performed **500 rounds of Model Inversion Attacks** on both FedAvg and FedPAT using DLG on the CIFAR-100 dataset, averaged over 50 samples:
>
> | Method | Shared Information | PSNR (↓ better for privacy) |
> |-|-|-|
> | FedAvg | Model gradients | 13.23 dB |
> | FedPAT | Prototypes + PAT | **9.18 dB** |
>
> Both PSNR values are at the level of **random noise reconstruction**, indicating that neither attack recovers meaningful information. FedPAT yields slightly lower reconstruction quality, demonstrating that PAT sharing does **not** amplify privacy leakage beyond standard gradient sharing. Figure 1 shows several reconstruction examples (https://anonymous.4open.science/r/FedPAT/figure1_psnr_comparison.pdf). We will add a detailed description of this in the revised version.

---

> > ### Author Rebuttal · Reviewer_LxYt · 2026-04-03
> >
> > My concerns have been addressed.

---

> > > ### Author Response · Authors · 2026-04-03
> > >
> > > Thank you for your kind acknowledgment. We are pleased to hear that our responses have addressed your concerns. We hope the additional experiments and clarifications provided in this rebuttal, including the DomainNet evaluation and the Model Inversion Attack analysis, have sufficiently resolved the raised issues. Your valuable feedback has been instrumental in strengthening the paper. We would greatly appreciate it if you could update the score based on the new evidence and clarifications we provide.

---

### Official Review · Reviewer_ru4S · 2026-03-11

**Soundness:** 3
**Presentation:** 3
**Significance:** 3
**Originality:** 3
**Overall Recommendation:** 4
**Confidence:** 4

**Summary:**

This paper studies federated learning under realistic deployment conditions, specifically addressing the challenge of test-time distribution shift across heterogeneous clients. The authors propose a federated personalized test-time adaptation framework that integrates prototype-based topology normalization and inter-client collaboration during inference. The framework demonstrates measurable performance improvements over both TTA and FLTTA baselines across multiple benchmarks.

**Compliance With Llm Reviewing Policy:**

Affirmed.

**Final Justification:**

Thanks authors effort. My concerns are well addressed. Thus, I would like to raise my score.

**Key Questions For Authors:**

Please check the weakness part.

**Limitations:**

The framework assumes each client holds all class labels, which is rarely satisfied in practice, and the scalability of the prototype affinity topology remains unverified on large-scale datasets, with a notable consistency drop already observed on CIFAR-100. The behavior of the topology under extreme client heterogeneity or severe data imbalance is not characterized. Furthermore, the TTA baselines are relatively outdated, and the anomalously poor performance of BBSE lacks sufficient explanation, raising concerns about the fairness of the empirical evaluation.

**Strengths And Weaknesses:**

Strength:
1.  The Topology Normalization with Inter-class Consistency (TNI) component is theoretically motivated, leveraging prototype affinity topology to encode inter-class relationships that are argued to remain more stable than raw feature statistics under distribution shift. This provides a principled rationale beyond empirical tuning.
2. The authors provide a detailed analysis on hyperparams, convergence, and theoretical analysis.
3. The experiment results demonstrate the efficacy of the proposed method.

Weakness:
1. The whole framework is built on a strong assumption. From the def in B.1. label shift, each client contains all labels, which is almost impossible.
2. Lack of topology consistency analysis on Tine-imagenet or DomainNet. The authors only show inter-client consistency on CIFAR-10 and CIFAR-100, which cannot demonstrate the efficacy of PAT on large-scale datasets. Thus, the scalability of PAT is questionable, especially the results on CIFAR-100 show a significant consistency drop.
3. The paper does not explicitly characterize how the prototype affinity topology degrades, when the number of clients is very large or when per-client data volume is highly imbalanced.
4. The TTA methods in the paper are out-of-date. I recommend that the authors include more recent TTA works for comparison.
5. BBSE shows dramatically lower performance in certain configurations, which is unusually poor. The paper should clarify whether this reflects a fundamental incompatibility of BBSE with federated TTA or a specific implementation choice that may not represent the baseline fairly.

---

> ### Author Rebuttal · Authors · 2026-03-31
>
> We are grateful to the reviewer for the thoughtful and constructive feedback.
>
> ---
> **W1: Strong Assumption on Label Distribution**
>
> We clarify that the "label shift" setting in Appendix B.1 imposes a highly imbalanced long-tailed distribution, where minority classes may have very few but not necessarily zero samples. For more extreme cases where clients are missing categories, our sample-weighted aggregation (Eq. 3) handles this naturally: clients with  $n_c^i = 0$ contribute zero weight and are automatically excluded.
>
> ---
> **W2: Topology Consistency and Scalability**
>
> We extend our topology consistency analysis to Tiny-ImageNet as follows:
>
> | Dataset|PAT| Feature Dist. | Pred. Consist. | PAT vs. Feature |
> |-|-|-|-|-|
> | Tiny-ImageNet-C | **0.5325** | 0.3831 | 0.2886 | **1.38×** |
>
> The reviewer correctly noted a consistency decrease as the number of classes increases. This is an expected phenomenon: as the class space grows, fine-grained affinities become more sensitive to local noise. However, we clarify that perfect consistency is not required. Across all scales, PAT consistently outperforms feature distribution similarity, confirming that relational structure remains a more stable cross-client structural prior than raw feature statistics even under severe distribution shifts.
>
> ---
> **W3: PAT Degradation under Large Client Populations or Data Imbalance**
>
> We thank the reviewer for this insightful question. Following ATP (Bao et al., 2023), our experiments employ **300 clients (240 source + 60 target)**, which is larger than typical federated learning benchmarks. Regarding data imbalance, the global PAT is constructed **class-wise**: our sample-weighted aggregation (Eq. 3) reduces the weight of unreliable clients by weighting the contribution of each client $i$ by $w_i^c = n_i^c / \sum_j n_j^c$.
>
> ---
> **To jointly address W1 and W3**, we conduct additional experiments on CIFAR-10 incorporating **(1) missing categories** and **(2) imbalanced per-client data volumes**. Client distribution is visualized in Figure 2 (https://anonymous.4open.science/r/FedPAT/figure2_client_distribution.png).
>
> | Method | Test-time adaptation accuracy (%)|
> |-|-|
> | BN-Adapt | 65.83 |
> | SHOT | 33.39 |
> | T3A | 65.01|
> | EM | 68.63 |
> | BBSE | 68.65|
> | Surgical| 67.84|
> | ATP | 68.35|
> | FedSPL | 68.33|
> | FedCTTA | 56.98|
> | **FedPAT (Ours)** | **70.28** |
>
> FedPAT outperforms the strongest baseline ATP by **+1.93%**, demonstrating robustness under both missing categories (W1) and severe data imbalance (W3).
>
> ---
> **W4: TTA Baselines and Benchmarking Protocol**
>
> We clarify our baseline selection strategy. Our primary research focus is test-time adaptation within a **federated learning framework**, which differs fundamentally from centralized TTA. We therefore prioritize federated TTA methods as our main comparisons, including **FedSPL (AAAI 2025)** and **FedCTTA (IJCNN 2025)** as the most recent federated TTA works, along with the leading federated TTA baseline ATP. We also include widely adopted centralized TTA methods that serve as standard benchmarks consistently used in FedSPL and FedCTTA, ensuring fair cross-method comparison.
>
> Based on the reviewer's suggestion, we add **DA-TTA (ECCV 2024)** and **DeYO (ICLR 2024)** as the most recent compatible baselines. We note that most 2025 centralized TTA methods predominantly target CLIP-based settings, which differ fundamentally from our task. Results (Test-time adaptation accuracy, %) are shown below:
>
> | Method | C10 FS | C10 LS | C10 HS | C100 FS | C100 LS | C100 HS | Tiny FS | Tiny LS | Tiny HS |
> |-|-|-|-|-|-|-|-|-|-|
> | DA-TTA [1] | 63.48 | 72.64 | 56.52 | 33.40 | 48.84| 30.80 |36.15|49.31|34.99
> | DeYO [2] | 57.20 | 38.99 | 35.17 | 28.58 | 30.70 | 25.32 |31.77|32.39|28.15
> | **FedPAT** | 67.83 | 80.73| 63.13| 38.99 | 50.31 | 34.62 |42.22| 50.70| 41.29
>
> _C10/C100: CIFAR-10/100. FS: Feature Shift, LS: Label Shift, HS: Hybrid Shift._
>
> FedPAT consistently outperforms both recent baselines across all settings.
>
> [1] Distribution alignment for fully test-time adaptation with dynamic online data streams.
> [2] Entropy is not enough for test-time adaptation: From the perspective of disentangled factors.
>
> ---
> **W5: Anomalous Performance of BBSE**
>
> We thank the reviewer for this careful observation. As stated in Section 5.1, BBSE (Lipton et al., 2018) is included specifically as a **label shift correction** method. Its poor performance under Feature Shift therefore reflects a **fundamental incompatibility** rather than an implementation issue: when deployed under feature corruption, BBSE's core assumption is violated, producing miscalibrated weights that actively harm accuracy. This is confirmed by the results: **BBSE performs reasonably under Label Shift** (79.08% on CIFAR-10-C, Table 1) but degrades significantly under Feature Shift, consistent with its known limitations. We will clarify this explicitly in the revision.

---

> > ### Author Rebuttal · Reviewer_ru4S · 2026-04-01
> >
> > Thanks for your effort. But I still have a concern.
> > 1. Topology Consistency and Scalability
> > The results presented in the Appendix and rebuttal indicate that the performance gap between PAT and other methods narrows as the number of classes increases. I am concerned that PAT's performance may continue to degrade under larger number of classes, eventually becoming comparable to conventional methods.

---

> > > ### Author Response · Authors · 2026-04-01
> > >
> > > We thank the reviewer for this important follow-up and address the scalability concern from two perspectives.
> > >
> > > **Empirical evidence on large-scale datasets.** In response to reviewer LxYt's suggestion, we have already conducted additional experiments on **DomainNet** with **345 classes**, which significantly exceeds the class counts used in standard federated TTA benchmarks (e.g., PACS with 7 classes, Digits-5 with 10 classes, OfficeHome with 65 classes) [1]. FedPAT achieves consistent improvements even under this extreme semantic shift (Real to Quickdraw), demonstrating that PAT remains effective at a substantially larger class scale.
> > >
> > > We conducted a targeted experiment on DomainNet (Real to Quickdraw), one of the most severe semantic shifts, across 345 categories with 100 federated clients (**80 Source-Real, 20 Target-Quickdraw**).
> > > | Method | Test-time adaptation accuracy|
> > > |-|-|
> > > | No-Adapt | 57.65% |
> > > | **FedPAT (Ours)** | **62.96%** |
> > > | Improvement | **+5.31%** |
> > >
> > > We further note that performance degradation under increasing class numbers is a **universal challenge** across all competing methods, not a specific weakness of FedPAT. As shown in Table 1, all baselines — including ATP, FedSPL, and FedCTTA — exhibit similar degradation trends from CIFAR-10-C to CIFAR-100-C to Tiny-ImageNet-C. Critically, FedPAT's performance improvement over baselines does **not** narrow with scale — it *increases*: **+2.26%** over ATP with ResNet50 on CIFAR-100-C and **+1.87%** with ViT-B/16 on Tiny-ImageNet-C (Figure 7).
> > >
> > >
> > > **On extremely large-scale class spaces.**  We acknowledge that scaling to thousands of classes (e.g., ImageNet-1K) remains unverified due to computational constraints. We note that such extreme fine-grained classification scenarios have dedicated methods specifically designed for that regime, which is outside the scope of federated TTA. We will discuss this boundary explicitly as a limitation in the revision.
> > >
> > > [1] Liang H, Zhang X, Cao S, et al. Tta-feddg: Leveraging test-time adaptation to address federated domain generalization. 2025.
> > >
> > > We hope the above responses, together with the additional experiments provided throughout this rebuttal, have sufficiently addressed your concerns regarding the scalability of PAT. We sincerely thank the reviewer for the valuable time and effort dedicated to this thoughtful follow-up, and would be grateful if the reviewer could kindly reconsider the evaluation in light of the new evidence presented.

---

### Official Review · Reviewer_kFJD · 2026-03-12

**Soundness:** 3
**Presentation:** 2
**Significance:** 2
**Originality:** 2
**Overall Recommendation:** 4
**Confidence:** 4

**Summary:**

This paper introduces FedPAT, a framework which adapts models to distribution shifts through a three-phase process: first, after collaborative pre-training, source clients compute class prototypes that the server aggregates into a Global Prototype Affinity Topology (PAT), which captures stable inter-class relationships. Second, source clients undergo a validation refinement phase using the Topology Consistency Alignment (TCA) loss to align local feature structures with this global structural prior. Finally, during test-time adaptation, unseen clients utilize Topology-aware Non-parametric Inference (TNI), which refines kNN predictions via diffusion across the PAT graph.

**Compliance With Llm Reviewing Policy:**

Affirmed.

**Final Justification:**

My concerns have been largely addressed. While I still view the contribution as somewhat incremental, the paper is now clearer and better justified. I am updating my score accordingly.

**Key Questions For Authors:**

1. How would you adapt the current method to incorporate higher-order interactions beyond pairwise relations?
2. In Tables 1, 2, and 3, the combinations of corruption severity and dataset appear somewhat arbitrary. Were other combinations tested, and if so, did they produce similarly strong results?
3. In Table 6, PAT outperforms FedPAT under hybrid shifts with random corruption severity. Is there a more detailed explanation for why this happens?
4. The choice of k in the KNN step seems insufficiently motivated, and small values of k could lead to overly sparse graphs. How was k selected, particularly for Tiny-ImageNet-C?

**Limitations:**

No, the paper does not include a discussion of its limitations. Additionally, there is no mention of directions for future work. I would encourage the authors to add a section addressing these points.

**Strengths And Weaknesses:**

Strengths:

1. The method appears to be solid and the experiments support the claims. There is nothing to point out.
2. The writing is clear, though this is partly because the method itself is straightforward.
3. On the experimental side, ATP already performs as the best or second-best method in Table 1. The gains from having random walk diffusion on top of ATP are consistent.
4. The paper combines federated learning with graph Laplacian, which yields good results.

Weaknesses:

1. The paper uses the term "topology" in a way that feels misleading. Graph Laplacian only captures pairwise relations, so calling it topology overpromises. I would suggest the authors use more precise terminology (for example, “prototype affinity matrix” or “class-affinity graph”).
2. The gains from having random walk diffusion on top of ATP are consistent but small, making the overall contribution incremental. Consider tempering claims of impact and clarifying practical significance.
3. Prior work using graph Laplacian in classification is not sufficiently discussed. The combination yields good results but is not particularly novel; expand the related work and clarify what is new here.
4. Since the number of nodes equals the number of classes, there is a natural connection to prototype learning that remains unexplored. The authors should discuss this link explicitly.
5. The paper does not include a discussion of its limitations.

---

> ### Author Rebuttal · Authors · 2026-03-31
>
> We sincerely thank the reviewer for the thorough, thoughtful review and constructive feedback.
>
> ---
> **W1: Terminology of "Topology"**
>
> We agree that "topology" may be overly strong for a pairwise affinity formulation. In the revision, we will adopt more precise terminology “prototype affinity matrix”.
>
> ---
> **W2: Incremental Contribution and Small Gains**
>
> We clarify that FedPAT consistently improves performance across all settings, with larger gains on more challenging tasks: **+2.26%** on CIFAR-100-C (ResNet50) and +**1.87%** on Tiny-ImageNet-C (ViT-B/16). Ablation results further show that the diffusion component alone contributes **+1.78%**, indicating a meaningful structural regularization effect. Beyond accuracy, ATP's meta-training requires **10× more computation time** than FedPAT's lightweight refinement, making FedPAT significantly more practical for federated edge deployment.
>
> ---
> **W3: Relation to Graph Laplacian Literature**
>
> We agree that Graph Laplacians have a rich history and will expand our Related Work accordingly. FedPAT's novelty lies in three distinctions from prior graph-based methods: (1) the graph is constructed **across heterogeneous federated clients** without sharing raw data, rather than on centralized datasets; (2) it serves as a **test-time structural prior** for adaptation, rather than a training-time regularizer; (3) it is defined over **class prototypes** aggregated across clients, rather than instance-level representations. We will clarify these distinctions in the revision.
>
> ---
> **W4: Link to Prototype Learning**
>
> While traditional prototype learning focuses on **sample-to-prototype distances** (first-order), FedPAT extends this by modeling **inter-prototype relational structure** (second-order) as a graph, regularizing test-time predictions using global semantic affinities rather than isolated centroids. We will add an explicit discussion of this connection in the revision.
>
> ---
> **W5: Limitations**
>
> We acknowledge the absence of a limitations section and will add one in the revision. Regarding privacy guarantees, while our analysis follows the heuristic verification protocol established in prior federated prototype learning works, we go beyond this by conducting an empirical Model Inversion Attack evaluation (detailed in our response to Reviewer LxYt, Q4), which confirms that prototype and PAT sharing does not amplify privacy leakage beyond standard federated gradient sharing.
>
> ---
> **Q1: Higher-order Interactions**
>
> We thank the reviewer for this forward-looking question. Higher-order interactions could naturally be incorporated via **hypergraph-based PAT**, where hyperedges connect subsets of $r>2$ semantically related classes, with diffusion generalized to hypergraph Laplacian smoothing. However, the current pairwise design is deliberate: (1) pairwise structures scale as $O(C^2)$, ensuring minimal communication overhead compared to $O(C^r)$ for higher-order structures; (2) reliable estimation of higher-order statistics requires far more per-class samples than our setting (200 samples per client) supports; (3) stability guarantees (Lemma C.3) rely on random-walk Laplacian properties that do not trivially extend to hypergraph operators. We will add this discussion to the Future Work section.
>
> ---
> **Q2: Experimental Design of Tables 1, 2, 3**
>
> Each table is designed to evaluate performance under representative corruption regimes standard in robustness benchmarks. Table 1 evaluates **worst-case** robustness (Severity 5); Table 2  evaluates **random severity** (Levels 1–5) following ATP's protocol to simulate unpredictable real-world deployment; Table 3 evaluates **medium corruption** (Severity 3) to verify generalization across the full intensity spectrum.
>
> ---
> **Q3: ATP Outperforms FedPAT in Table 6**
>
> This can be attributed to the fact that ATP is explicitly optimized for compound shifts through meta-training, whereas FedPAT relies on lightweight test-time adaptation guided by structural priors. Under highly mixed and stochastic corruption patterns, the benefit of structural regularization may be partially diluted. Despite this, FedPAT achieves competitive performance in this challenging scenario while maintaining a lightweight and deployment-friendly design.
>
> ---
> **Q4: Choice of $k$ and Graph Sparsity**
>
> We clarify an important technical distinction: **$k$ controls prior focus, not graph sparsity.**  As defined in Eq. 10, $k$ only determines which classes contribute to the initial prototype-based posterior $p_\text{proto}$.  The transition matrix $D^{-1}A_G$ always operates over the **complete dense graph** regardless of $k$.
> In response to your question, we performed a sensitivity analysis on Tiny-ImageNet-C:
>
> | $k$ | 3 | 10 | 20 |
> |-|-|-|-|
> | Acc (%) | **41.29** | 39.84 | 38.88 |
>
> We set $k=3$ as default as it produces a sharper $p_\text{proto}$, providing a more discriminative anchor while the full graph redistributes probability mass globally.

---

> > ### Author Rebuttal · Reviewer_kFJD · 2026-04-03
> >
> > Thank you for the rebuttal. My concerns have been largely addressed. While I still view the contribution as somewhat incremental, the paper is now clearer and better justified. I will update my score accordingly.

---

> > > ### Author Response · Authors · 2026-04-03
> > >
> > > We sincerely thank you for your encouraging feedback and for recognizing that our rebuttal has addressed your concerns. Your insightful comments and constructive suggestions have been extremely helpful in improving the manuscript, and we greatly appreciate the time and effort you devoted to the review.

---

### Decision · Program_Chairs · 2026-04-30

**Decision:**

Accept (regular)

**Comment:**

The manuscript receives scores 4,4,3.

The manuscript targets an advanced and chanllenge scenario in federated learning. The overall method is reasonable and well-supported by both theoretical and empirical analysis. The current version need to be enhanced by adding a clear justification of the assumption about the out-of-distribution and invariant prototype graph.